# The Arabidopsis active demethylase ROS1 *cis*-regulates defence genes by erasing DNA methylation at promoter-regulatory regions

**Thierry Halter, Jingyu Wang, Delase Amesefe, Emmanuelle Lastrucci, Magali Charvin, Meenu Singla Rastogi, Lionel Navarro\***

Institut de Biologie de l'Ecole Normale Supérieure (IBENS), Centre National de la Recherche Scientifique (CNRS), Institut National de la Santé et de la Recherche Médicale (INSERM), Université de recherche Paris, Sciences & Lettres (PSL), Paris, France

**Abstract** Active DNA demethylation has emerged as an important regulatory process of plant and mammalian immunity. However, very little is known about the mechanisms by which active demethylation controls transcriptional immune reprogramming and disease resistance. Here, we first show that the Arabidopsis active demethylase ROS1 promotes basal resistance towards *Pseudomonas syringae* by antagonizing RNA-directed DNA methylation (RdDM). Furthermore, we demonstrate that ROS1 facilitates the flagellin-triggered induction of the disease resistance gene *RMG1* by limiting RdDM at the 3' boundary of a transposable element (TE)-derived repeat embedded in its promoter. We further identify flagellin-responsive ROS1 putative primary targets and show that at a subset of promoters, ROS1 erases methylation at discrete regions exhibiting WRKY transcription factors (TFs) binding. In particular, we demonstrate that ROS1 removes methylation at the orphan immune receptor *RLP43* promoter, to ensure DNA binding of WRKY TFs. Finally, we show that ROS1-directed demethylation of *RMG1* and *RLP43* promoters is causal for both flagellin responsiveness of these genes and for basal resistance. Overall, these findings significantly advance our understanding of how active demethylases shape transcriptional immune reprogramming to enable antibacterial resistance.

**\*For correspondence:**
lionel.navarro@ens.psl.eu

**Competing interests:** The authors declare that no competing interests exist.

## Introduction

Plants are permanently exposed to microbes including pathogens and rely on potent active immune responses to control infections. The first layer of the plant immune system involves the recognition of pathogen- or microbe-associated molecular patterns (PAMPs or MAMPs), which are sensed by surface-localized pattern-recognition receptors (PRRs) (*Couto and Zipfel, 2016*). Plant PRRs are composed of receptor-like kinases (RLKs) and receptor-like proteins (RLPs), which are structurally and functionally analogous to animal Toll-like receptors (TLRs) (*Boutrot and Zipfel, 2017*). Flagellin Sensing 2 (FLS2) is a well-characterized RLK PRR, which recognizes a conserved 22 amino acids epitope from the N-terminal part of the bacterial flagellin, named flg22 (*Boller and Felix, 2009*). Upon ligand binding, FLS2 initiates a complex phosphorylation cascade at the PRR complex that leads to early signalling events, which include production of reactive oxygen species (ROS), activation of mitogen-activated protein kinases (MAPKs) and differential expression of thousands of genes, which are in part regulated by WRKY transcription factors (TFs) (*Navarro et al., 2004*; *Zipfel et al., 2004*; *Birkenbihl et al., 2017*). To enable disease, pathogens secrete effectors that suppress PTI. For instance, the Gram-negative hemibiotrophic pathogenic bacterium *Pseudomonas syringae* pv.

*tomato* strain DC3000 (*Pto* DC3000) injects 36 type-III effectors into plant cells to dampen PTI (*Wei et al., 2015*). Plants have evolved disease resistance (R) proteins that can perceive the presence of pathogen effectors and trigger a host counter-counter defence (*Monteiro and Nishimura, 2018*). Most R proteins belong to the nucleotide binding (NB) leucine-rich repeat (LRR) NLR superfamily, which is also present in animals (*Jones et al., 2016*). Plant NLRs contain N-terminal coiled-coil (CC) or Toll-interleukine (TIR) domains, a central NB domain and a C-terminal LRR, and are thus referred to as CNLs or TNLs, respectively (*Monteiro and Nishimura, 2018*). These immune receptors can sense, directly or indirectly, pathogen effectors and mount effector-triggered immunity (ETI), a potent immune response that significantly overlaps with PTI, although with a stronger amplitude (*Jones et al., 2016*; *Thomma et al., 2011*). However, very little is known about the functional relevance of *NLRs* in PTI and/or basal resistance towards phytopathogens (*Roth et al., 2017*).

Rapid and robust activation of plant immune responses is crucial to limit the spread and multiplication of phytopathogens. On the other hand, their sustained induction often leads to cell death and developmental defects. For instance, the constitutive expression of *NLRs* often triggers ectopic cell death (*Oldroyd and Staskawicz, 1998*; *Zhang et al., 2004*; *Swiderski et al., 2009*; *Bernoux et al., 2011*; *Nishimura et al., 2017*). A tight control of both plant immune receptors and downstream signalling factors is therefore crucial and involves multi-layered transcriptional, post-transcriptional and post-translational regulatory mechanisms (*Zipfel et al., 2004*; *Halter and Navarro, 2015*; *van Wersch et al., 2020*; *Deleris et al., 2016*). The transcriptional control of defence genes by DNA methylation and demethylation has emerged as a central regulatory process of the plant immune system (*López et al., 2011*; *Yu et al., 2013*; *Dowen et al., 2012*; *López Sánchez et al., 2016*; *Le et al., 2014*; *Deleris et al., 2016*; *Kong et al., 2020*).

DNA methylation is an epigenetic mark that negatively regulates the transcription and transposition of transposable elements (TEs). It can also trigger transcriptional silencing of some genes carrying TEs/repeats in their vicinity (*Matzke and Mosher, 2014*). Homeostasis of DNA methylation relies on the equilibrium between methylation and active demethylation pathways, and has been extensively studied in *Arabidopsis thaliana*. In this model organism, DNA methylation is established by the RNA-directed DNA methylation (RdDM) pathway, which is directed by short interfering RNAs (siRNAs) (*Matzke and Mosher, 2014*). These siRNAs are mainly produced from PolIV-dependent RNAs (P4RNAs) that are converted into double-stranded RNAs (dsRNAs) by RNA-DEPENDENT RNA POLYMERASE 2 (RDR2) (*Matzke and Mosher, 2014*; *Blevins et al., 2015*; *Zhai et al., 2015*; *Yang et al., 2016*). The resulting dsRNAs are predominantly processed by DICER-LIKE 3 (DCL3) into 23–24 siRNAs, but additionally, in some instances, by DCL2 and DCL4 into 22 nt and 21 nt siRNAs, respectively, which are all competent for RdDM (*Xie et al., 2004*; *Matzke and Mosher, 2014*; *Panda et al., 2020*). These siRNAs further direct ARGONAUTE 4 (AGO4) to TEs/repeats through base pairing with transcripts generated by Pol V (*Zilberman et al., 2004*; *Chan et al., 2004*; *Qi et al., 2006*; *Wierzbicki et al., 2009*). AGO4 then binds to DNA targets and recruits the *de novo* methyltransferase DOMAIN REARRANGED METHYLTRANSFERASE 2 (DRM2) that catalyses methylation in all cytosine sequence contexts (CG, CHG and CHH, where H is any nucleotide but not G) (*Cao and Jacobsen, 2002*; *Cao et al., 2003*; *Lahmy et al., 2016*). During DNA replication, symmetric CG and CHG methylations are maintained by METHYLTRANSFERASE 1 (MET1) and CHROMOMETHYLASE 3 (CMT3), respectively, while asymmetric CHH methylation is either actively perpetuated by RdDM or maintained by a siRNA-independent process mediated by CMT2 (*Cao and Jacobsen, 2002*; *Matzke and Mosher, 2014*; *Stroud et al., 2014*). On the other hand, Arabidopsis encodes four active demethylases with 5-methylcytosine DNA glycosylase/lyase activities, namely DEMETER (DME), DME-Like 1 (DML1)/ROS1 (Repressor of Silencing 1), DML2 and DML3 (*Zhang et al., 2018*). Both ROS1 and DME actively remove DNA methylation in all methylated cytosine contexts through a base excision repair (BER) mechanism (*Gong et al., 2002*; *Mok et al., 2010*). ROS1 is the active demethylase that has been the most characterized in vegetative tissues (*Zhang et al., 2018*). Mechanistically, ROS1 prunes siRNA-dependent or -independent DNA methylation at thousands of loci and this process often limits the spreading of DNA methylation at TE/repeat boundaries (*Tang et al., 2016*). Importantly, ROS1-directed removal of DNA methylation at specific promoters is critical to ensure a proper expression of genes required for developmental or abiotic stress responses (*Yamamuro et al., 2014*; *Gong et al., 2002*; *Kim et al., 2019*).

Several studies have unveiled a major role for DNA methylation in susceptibility against non-viral phytopathogens. For examples, DNA methylation-defective mutants of Arabidopsis are more

resistant to *Pto* DC3000 and the obligate biotrophic oomycete pathogen *Hyaloperonospora arabidopsidis* (*Pavet et al., 2006*; *Yu et al., 2013*; *Dowen et al., 2012*; *López Sánchez et al., 2016*). Conversely, *ros1* mutants display enhanced susceptibility towards *Pto* DC3000 and *Hyaloperonospora arabidopsidis* (*Yu et al., 2013*; *López Sánchez et al., 2016*), indicating that ROS1 promotes basal resistance against those pathogens. Furthermore, ROS1, DML2, DML3, and DME act cooperatively to orchestrate resistance towards *Fusarium oxysporum*, a devastating hemibiotrophic vascular fungal pathogen infecting a wide range of economically important crops (*Le et al., 2014*; *Schumann et al., 2019*). Altogether, these studies suggest that immune-responsive genes are likely to be regulated by DNA methylation and/or demethylation, which has been demonstrated at a subset of pathogen-responsive genes (*Yu et al., 2013*; *Le et al., 2014*; *López Sánchez et al., 2016*; *Schumann et al., 2019*; *Kong et al., 2020*). These findings also suggest that demethylation of defence gene promoters might ensure the DNA/chromatin binding of TFs during pathogen infection and/or elicitation, although this hypothesis has never been tested experimentally.

Here, we have characterized the Arabidopsis demethylase ROS1 in the context of antibacterial immunity. We first demonstrate that ROS1 positively regulates basal resistance against *Pto* DC3000 by antagonizing RdDM activity. Consistent with this observation, we find that the *TNL RESISTANCE METHYLATED GENE 1* (*RMG1*) contributes to basal resistance against *Pto* DC3000, and that ROS1-directed suppression of RdDM at *RMG1* promoter ensures a proper induction of this gene during PTI. Furthermore, we retrieve the whole set of flg22-responsive genes that are controlled by ROS1 and show that PAMP-triggered inducibility at two ROS1 targets, namely *RMG1* and *RLP43,* depends on the *cis*-effect of demethylation at their promoters. Furthermore, ROS1-directed demethylation at a specific sequence of the *RLP43* promoter, which carries a functional 'W-box' WRKY-binding site, is critical for the DNA binding of PAMP-responsive WRKY TFs. Overall, this study reveals the extent to which ROS1 orchestrates Arabidopsis transcriptional immune reprogramming and unveils a crucial role for this demethylase in the regulation of WRKY-DNA binding at immune-responsive promoters.

## Results

### Arabidopsis *ROS1* promotes resistance towards *Pto* DC3000, mostly by antagonizing *DCL2* and/or *DCL3* functions

We have previously reported that *ros1* mutants exhibit enhanced spreading of *Pto* DC3000 in Arabidopsis leaf secondary veins (*Yu et al., 2013*). This phenotype was further confirmed here when *ros1-3* and *ros1-4* mutant leaves were wound-inoculated with a GFP-tagged *Pto* DC3000 (*Pto* DC3000-GFP) (*Figure 1A and B*). In addition, an enhanced bacterial titre was observed in these *ros1* alleles dip-inoculated with *Pto* DC3000-GFP (*Figure 1C*). These data indicate that ROS1 positively regulates basal resistance against *Pto* DC3000. It has been shown that ROS1 antagonizes methylation at thousands of loci, which are methylated in a RdDM-dependent or -independent manner (*Tang et al., 2016*). To determine whether ROS1 could promote basal resistance by counteracting RdDM, we have performed the above assays in a *ros1-3 dcl2-1 dcl3-1* triple mutant (*ros1dcl23*), in which the biogenesis of 22 to 24 nt siRNAs is abolished. Interestingly, we found that both the enhanced vascular propagation and apoplastic growth of *Pto* DC3000-GFP detected in the *ros1-3* mutant were reduced in *ros1dcl23* mutants, and almost comparable to the phenotypes of Col-0-infected plants (*Figure 1D–F*). These data indicate that DCL2 and/or DCL3 are mainly responsible for the enhanced susceptible phenotypes observed in *ros1-3*-infected mutants. They also suggest that some defence genes are likely hypermethylated and silenced in the *ros1-3* mutants through the action of, at least in part, DCL2- and/or DCL3-dependent siRNAs.

### The ROS1 target *RMG1* is a functional disease resistance gene that contributes to basal resistance towards *Pto* DC3000

ROS1 has previously been shown to ensure a proper flg22-triggered induction of *RMG1*, an orphan *TNL* that is demethylated by ROS1 in its promoter (*Yu et al., 2013*). As a result, a strong reduction in flg22-mediated inducibility of *RMG1* is observed in the absence of ROS1 (*Figure 2D and F*; *Yu et al., 2013*). To investigate the possible contribution of *RMG1* in basal resistance against *Pto* DC3000, we have isolated and characterized two independent T-DNA insertion lines, which lack *RMG1* mRNA in leaves treated with flg22 compared to Col-0 (*Figure 2A*, *Figure 2—figure*

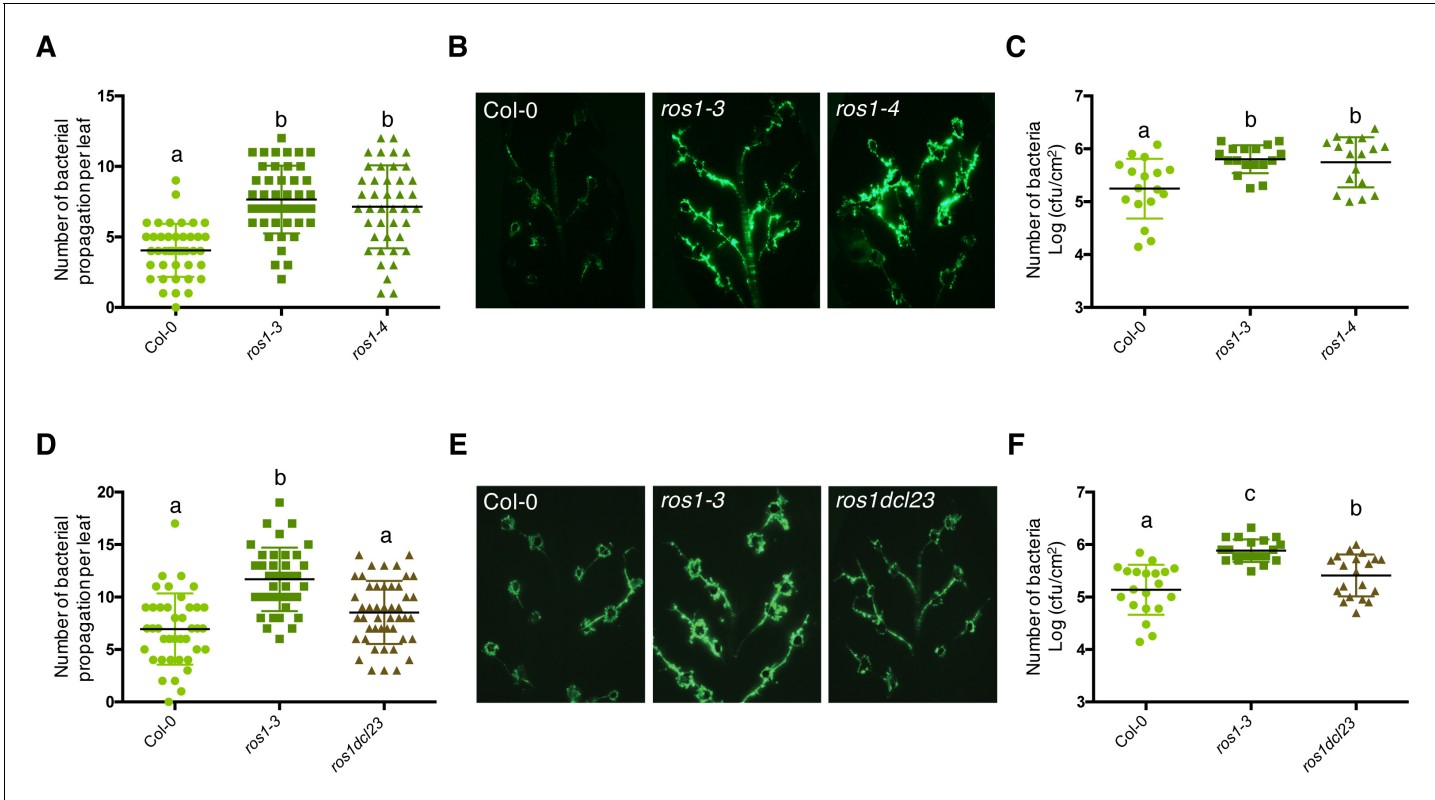

**Figure 1.** The enhanced *Pto* DC3000 disease susceptibility observed in *ros1*-infected mutants is mainly dependent on *DCL2* and/or *DCL3* functions. (**A**) Increased *Pto* DC3000 vascular propagation in the two independent *ros1* mutant alleles, *ros1-3* and *ros1-4*. Secondary veins of 5-week-old Col-0, *ros1-3* and *ros1-4* mutants were inoculated with a virulent GFP-tagged *Pto* DC3000 strain (*Pto* DC3000-GFP) at $5 \times 10^6$ cfu ml$^{-1}$ using the toothpick inoculation method. Inoculation was done on six secondary veins per leaf and two sites of inoculation per vein. At least 10 leaves per experiment were quantified. The number of *Pto* DC3000-GFP spreading events from the wound inoculation sites was quantified after 3 days under UV light using a macrozoom. When the bacteria propagated away from any of the 12 inoculation sites, it was indexed as propagation with a possibility of maximum 18 propagations per leaf. The values from three independent experiments were considered for the comparative analysis. Statistical significance was assessed using a one-way ANOVA test and Tukey's multiple comparisons test. (**B**) Representative pictures of the GFP fluorescence observed at the whole leaf level on the plants depicted in **A**. (**C**) Enhanced *Pto* DC3000 apoplastic growth in the two independent *ros1* mutant alleles, *ros1-3* and *ros1-4*. Five-week-old plants of Col-0, *ros1-3,* and *ros1-4* mutants were dip-inoculated with *Pto* DC3000-GFP at $5 \times 10^7$ cfu ml$^{-1}$. Bacterial titres were monitored at 3 days post-inoculation (dpi). Each data point represents bacterial titre at four leaf discs extracted from a single leaf. Two leaves out of three plants per line per experiment, and from three independent experiments were considered for the comparative analysis. Statistical significance was assessed using a one-way ANOVA test and Tukey's multiple comparisons test. (**D**) Increased bacterial propagation in the vein observed in *ros1-3* is rescued in the *ros1dcl23* triple mutant. Secondary veins of 5-week-old Col-0, *ros1-3* and the triple *ros1dcl23* mutants were inoculated as in **A** and the results analysed as in **A**. (**E**) Representative pictures of the GFP fluorescence observed at the whole leaf level on the plants presented in **D**. (**F**) Enhanced *Pto* DC3000 apoplastic growth in *ros1* is partially rescued in the *ros1dcl23* background. Five-week-old plants of Col-0, *ros1-3*, and *ros1dcl23* were inoculated as in **C** and the results were analysed as in **C**.

The online version of this article includes the following source data for figure 1:

**Source data 1.** Original data of bacterial propagation assays for *Figure 1A,C,D, and F*.

*supplement 1*). Both mutants exhibited elevated bacterial vascular propagation and apoplastic growth compared to Col-0-infected plants (*Figure 2B and C*), indicating that *RMG1* is a functional disease resistance gene that contributes to basal resistance towards *Pto* DC3000. These data also suggest that the heightened susceptibility of the *ros1*-infected mutants might be in part attributed to the silencing of *RMG1*.

## ROS1 limits the spreading of DNA methylation at the 3' boundary of a remnant RC/Helitron TE, which is embedded in the *RMG1* promoter

*RMG1* contains two remnant RC/Helitron TEs in its promoter: the distal repeat *AtREP4* (*At4TE29275*) and the proximal repeat *AtREP11* (*At4TE29280*) (*Figure 2D*). *AtREP4* is targeted by

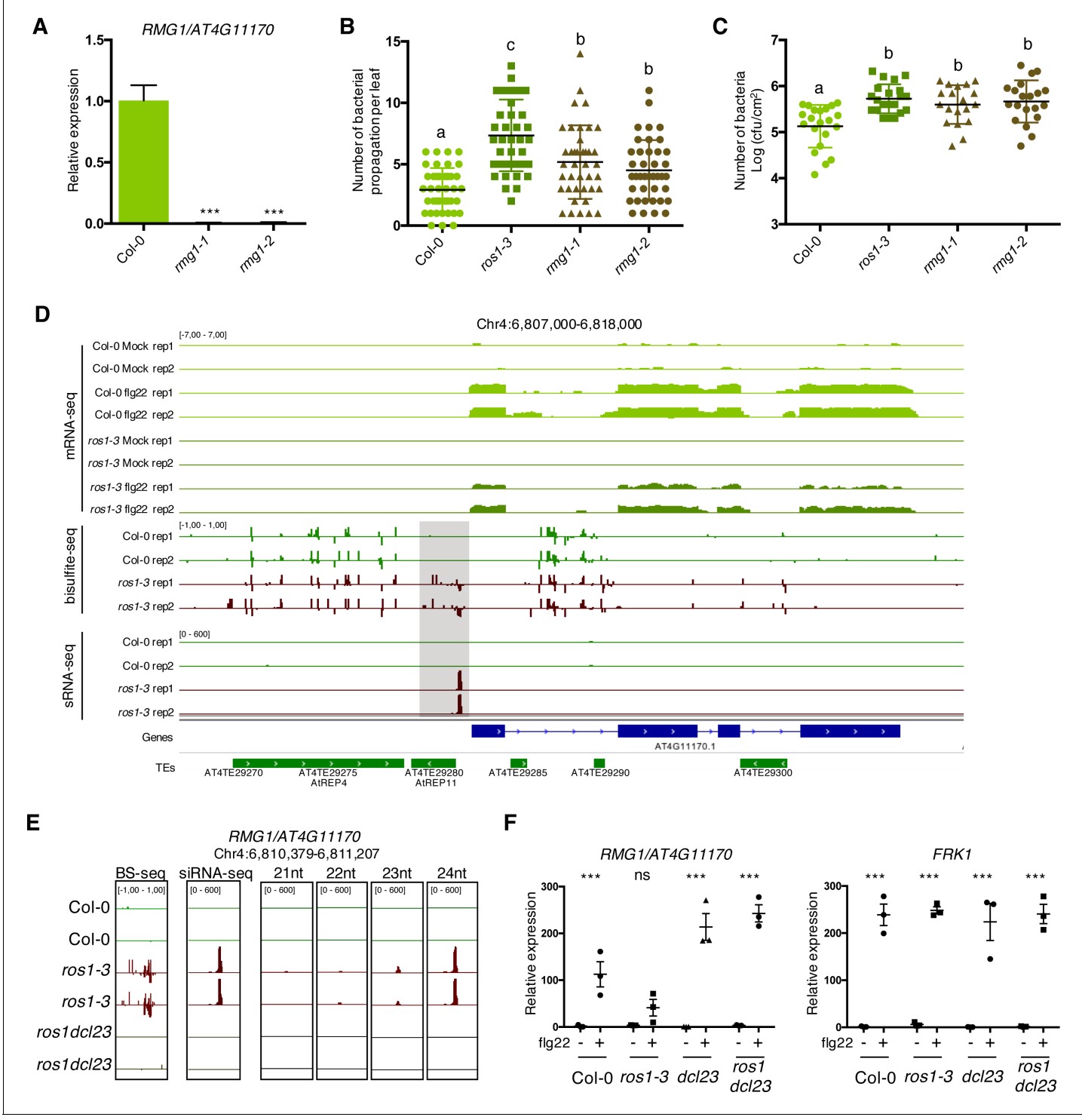

**Figure 2.** The *TNL* gene *RMG1* contributes to apoplastic and vascular resistance against *Pto* DC3000 and its flg22-triggered induction is negatively regulated by RNA-directed DNA methylation (RdDM) in the absence of ROS1. (**A**) *RMG1* mRNA levels in Col-0 and the two *rmg1* mutant alleles, namely *rmg1-1* and *rmg1-2*, were monitored by RT-qPCR at 6 hr after syringe-infiltration of mock (water) or 1 µM of flg22 peptide. (**B**) *RMG1* positively regulates vascular resistance towards *Pto* DC3000. Secondary veins of 5-week-old Col-0, *ros1-3*, *rmg1-1*, and *rmg1-2* plants were inoculated with *Pto* DC3000-GFP at $5 \times 10^6$ cfu ml$^{-1}$ using the toothpick inoculation method. Inoculation was done on six secondary veins per leaf and two sites of inoculation per vein. At least 10 leaves per condition were quantified. The number of *Pto* DC3000-GFP spreading events from the wound inoculation sites was quantified after 3 days under UV light using a macrozoom. When the *Pto* DC3000-GFP propagated away from any of the 12 inoculation sites, it was indexed as propagation with a possibility of maximum 18 propagations per leaf. The values from three independent experiments were

*Figure 2 continued on next page*

Figure 2 continued

considered for the comparative analysis. Statistical significance was assessed using a one-way ANOVA test. (C) *RMG1* positively regulates apoplastic resistance towards *Pto* DC3000. Five-week-old Col-0, *ros1-3*, *rmg1-1*, and *rmg1-2* plants were dip-inoculated with *Pto* DC3000-GFP at $5 \times 10^7$ cfu ml$^{-1}$. Bacterial titres were monitored at 3 days post-infection (dpi). Each data point represents bacterial titre at four leaf discs extracted from two different leaves. At least three leaves out of four plants per line per experiment and from three independent experiments were considered for the comparative analysis. Statistical significance was assessed using a one-way ANOVA test. (D) Flg22-triggered induction of *RMG1* is compromised in *ros1-3*-elicited mutant and correlates with an increased DNA methylation and siRNA levels at the remnant RC/Helitron TE *AtREP11*, and particularly at its 3′ boundary. IGV snapshots showing mRNA levels (mRNA-seq) after syringe-infiltration of mock (water) or 1 μM of flg22 peptide for Col-0 and *ros1-3*, and cytosine DNA methylation levels (Bs-Seq) and siRNA levels (sRNA-seq) in 5-week-old untreated rosette leaves of Col-0 and *ros1-3*, at the *RMG1* locus. The hyperDMR is highlighted by the dotted box. (E) Levels of different siRNA species at the 3′ boundary of *AtREP11*. IGV snapshots representing the levels of methylation (BS-seq), total siRNA species (siRNA-seq), and different size of siRNA species (21nt, 22nt, 23nt, and 24nt siRNAs) in 5-week-old rosette leaves of Col-0, *ros1-3*, and *ros1dcl23*. (F) The flg22-triggered induction of *RMG1* is fully restored in *ros1dcl23*-elicited triple mutants. RT-qPCR analysis depicting *RMG1* and *FRK1* mRNA levels in Col-0, *ros1-3*, *dcl23*, and *ros1dcl23* 5-week-old rosette leaves treated with either mock (water) or 1 μM of flg22 for 6 hr. The mRNA levels are relative to the level of *UBQ* transcripts. Statistical significance of flg22 treatment on expression was assessed using a two-way ANOVA test and a Sidak's multiple comparisons test. Asterisks indicate statistical significance (*: p<0.05, **: p<0.01, ***: p<0.001, ns: not significant).

The online version of this article includes the following source data and figure supplement(s) for figure 2:

**Source data 1.** Original qRT-PCR data for *Figure 2A and F*, and bacterial propagation data for *Figure 2B and C*.

**Figure supplement 1.** Scheme representing T-DNA insertion sites at the *RMG1* locus.

**Figure supplement 2.** *FRK1* and *bZIP60* promoters are not hypermethylated in *ros1-3* mutants.

**Figure supplement 3.** The flg22-triggered induction of *RMG1* and *RBA1* is restored, while *RLP43* remains in a repressed state, in the *ros1dcl23*-elicited mutant.

**Figure supplement 3—source data 1.** Original qRT-PCR data for *Figure 2—figure supplement 3*.

23–24 nt siRNAs and methylated in the Arabidopsis reference accession Col-0, a regulatory process that presumably maintains a low basal expression of this gene in untreated conditions (*Yu et al., 2013*). By contrast, *AtREP11* (*At4TE29280*), which is located 216 bp upstream of *RMG1* start codon, is unmethylated in Col-0 but hypermethylated in a *ros1* mutant, as previously demonstrated by a targeted bisulfite Sanger sequencing approach (*Yu et al., 2013*). These results were confirmed by using a whole-genome bisulfite sequencing (BS-seq) approach in rosette leaves of untreated Col-0 and *ros1-3* mutants (*Figure 2D*). We found that the hyper differentially methylated region (hyperDMR) detected at the *AtREP11* repeat was particularly pronounced at the 3′ boundary of this remnant RC/ Helitron TE in the *ros1-3* mutant, while this region was unmethylated in Col-0 plants (*Figure 2D*). This result is consistent with a role of ROS1 in limiting the spreading of DNA methylation at TE boundaries (*Tang et al., 2016*). To determine whether such hyperDMR could be directed by siRNAs, we monitored the accumulation of siRNAs at the *RMG1* promoter by analysing small RNA sequencing (sRNA-seq) datasets generated in rosette leaves of untreated Col-0, *ros1-3* and *ros1dcl23* mutants. While no siRNA was retrieved at the 3′ boundary of *AtREP11* in Col-0 plants, we detected the accumulation of siRNA species of different sizes in the *ros1-3* mutant, ranging from low levels in the 21–22 nt species to high levels in the 23–24 nt species (*Figure 2D and E*). All these siRNAs were no longer produced in *ros1dcl23* mutants, a molecular effect which was associated with an absence of cytosine methylation at this ROS1-targeted region (*Figure 2E*). Collectively, these data indicate that DCL2- and/or DCL3-dependent siRNAs direct RdDM predominantly at the 3′ boundary of *AtREP11* in the *ros1-3* mutant. They also suggest that, in Col-0 plants, ROS1 counteracts the biogenesis of DCL2- and DCL3-dependent siRNAs at this *RMG1* promoter region to restrict DNA methylation spreading at the 3′ boundary of *AtREP11*.

## ROS1-directed suppression of siRNA biogenesis at the *RMG1* promoter is required for a proper induction of this gene during flg22 elicitation

We next assessed whether ROS1-directed suppression of RdDM at the *AtREP11* and its 3′ boundary could be required for the proper induction of *RMG1* during flg22 elicitation. For this end, we challenged Col-0, *ros1-3*, *dcl2-1 dcl3-1* (*dcl23*) and *ros1dcl23* plants with either mock or flg22 for 6 hr and further monitored the mRNA accumulation of *RMG1*. We found a compromised induction of *RMG1* in the *ros1-3*-elicited mutant that was not observed for *Flg22-induced Receptor-like Kinase 1* (*FRK1*), which is not hypermethylated in *ros1-3* mutants, and thus served as a control (*Figure 2F*, *Figure 2—figure supplement 2*, *Figure 2—figure supplement 3*). By contrast, a full restoration of

flg22-triggered inducibility of *RMG1* was observed in the *ros1dcl23*-elicited mutant (*Figure 2F*, *Figure 2—figure supplement 3*). These data indicate that RdDM of *AtREP11* and its 3' boundary, which is specifically detected in *ros1-3* mutants, negatively regulates the flg22 inducibility of *RMG1*. They also suggest that ROS1-directed demethylation of this *RMG1* promoter region is crucial to ensure a proper transcriptional activation of this gene during flg22 elicitation.

## ROS1 directs promoter demethylation of flg22-induced ROS1 targets, mostly by antagonizing *DCL2* and/or *DCL3* functions

To identify the whole set of immune-responsive genes that are regulated by ROS1, we further used a RNA sequencing (RNA-seq) method and applied a statistical analysis that retrieved transcripts that are differentially accumulated at 6 hr after flg22 treatment in Col-0 *versus ros1-3* plants. Using this approach, we identified 2076 differentially expressed genes, among which, 907 were less-induced and 1169 less-repressed in flg22-treated *ros1-3* mutants compared to Col-0 (*Figure 3A*). To identify ROS1 putative primary targets, we next determined which of the 2943 hyperDMRs identified in untreated *ros1-3* plants were present in an interval covering the gene-body plus 2 Kb upstream and downstream of the 2076 flg22-sensitive *ROS1* targets. As a result of this analysis, 217 candidate genes were recovered, representing ~10% of the whole flg22-responsive and ROS1 regulated genes. Among them, 102 less-induced (promoter regions: 44 DMRs; gene bodies: 33; downstream regions: 25) and 115 less-repressed (promoter regions: 67 DMRs; gene bodies: 27; downstream regions: 21) genes were recovered (*Figure 3A*, *Supplementary File 1*). To gain more insights into the regulatory function of ROS1 during PTI, we focused subsequent analyses on flg22-induced ROS1 putative primary targets (*Figure 3B*, *Figure 3—figure supplement 1*). We found that the overall increase of methylation levels at the 102 less-induced genes and at the whole 2943 ROS1 targets, in *ros1-3* compared to Col-0 plants, was almost exclusively observed in 2 Kb upstream sequence regions (*Figure 3C*). An increase in siRNA accumulation was also detected at both the promoters of the flg22-induced and overall ROS1 targets in *ros1-3 versus* Col-0 plants (*Figure 3D*, *Figure 3—figure supplement 2*). By contrast, the overall enhanced DNA methylation and siRNA levels observed in *ros1-3* mutants at these loci were almost restored to Col-0 levels in *ros1dcl23* mutants (*Figure 3C and D*, *Figure 3—figure supplement 2*). The latter data suggest that ROS1 directs promoter demethylation mostly by antagonizing *DCL2* and/or *DCL3* functions.

## ROS1 facilitates the flg22-triggered induction of three distinct targets by preventing hypermethylation at discrete regions in their promoters

We next selected, from our RNA-seq datasets, candidate genes that were strongly induced by flg22 in Col-0 plants and showing pronounced compromised flg22 inducibility in elicited *ros1-3* mutants (*Figure 3B*, *Figure 3—figure supplement 1*). These genes include the TIR-only resistance gene *RBA1* (*AT1G47370*), which was previously shown to be repressed by DNA methylation in Col-0 and to recognize the bacterial effector HopBA1 in the *Arabidopsis thaliana* Ag-0 accession (*Nishimura et al., 2017*), the orphan *receptor-like protein 43* (*RLP43*) (*AT3G28890*), which presents typical features of RLP PRRs (*Steidele and Stam, 2020*), and a phosphoglycerate mutase gene (*AT1G08940*) (*Figure 3B*). We further monitored their mRNA accumulation in Col-0, *ros1-3*, *dcl23* and *ros1dcl23* challenged with either mock or flg22 for 6 hr. The expression pattern of the phosphoglycerate mutase gene was similar to the one of *RMG1*: the compromised flg22 induction of this gene was almost fully rescued in elicited *ros1dcl23* mutants, which is consistent with a loss of siRNAs and DNA methylation at its hyperDMR in untreated *ros1dcl23* mutants (*Figure 3E and G*). The flg22-triggered induction of *RBA1* was restored in *ros1dcl23* to levels similar to *dcl23* mutants, but not to the same extent as in elicited Col-0 plants (*Figure 3G*, *Figure 2—figure supplement 3*). Furthermore, this effect was associated with a partial decrease in *RBA1* promoter methylation at the *RBA1* hyperDMR, in *ros1dcl23* compared to *ros1-3* mutants (*Figure 3E*). By contrast, *RLP43* remained fully silenced in the *ros1dcl23*-treated mutants, as in elicited *ros1-3* mutants, which is consistent with comparable levels of *RLP43* promoter methylation in both untreated mutants (*Figure 2—figure supplement 3*, *Figure 3E*, *Figure 3G*). It is noteworthy that the unaltered *RLP43* hyperDMR detected in the *ros1dcl23* mutants was associated with a moderate remaining accumulation of siRNAs, which was shifted from 23 to 24 nt siRNAs in *ros1-3* to 21 nt siRNAs in *ros1dcl23* mutants (*Figure 3E*, *Figure 3—figure supplement 3*). It was also accompanied by the presence of longer

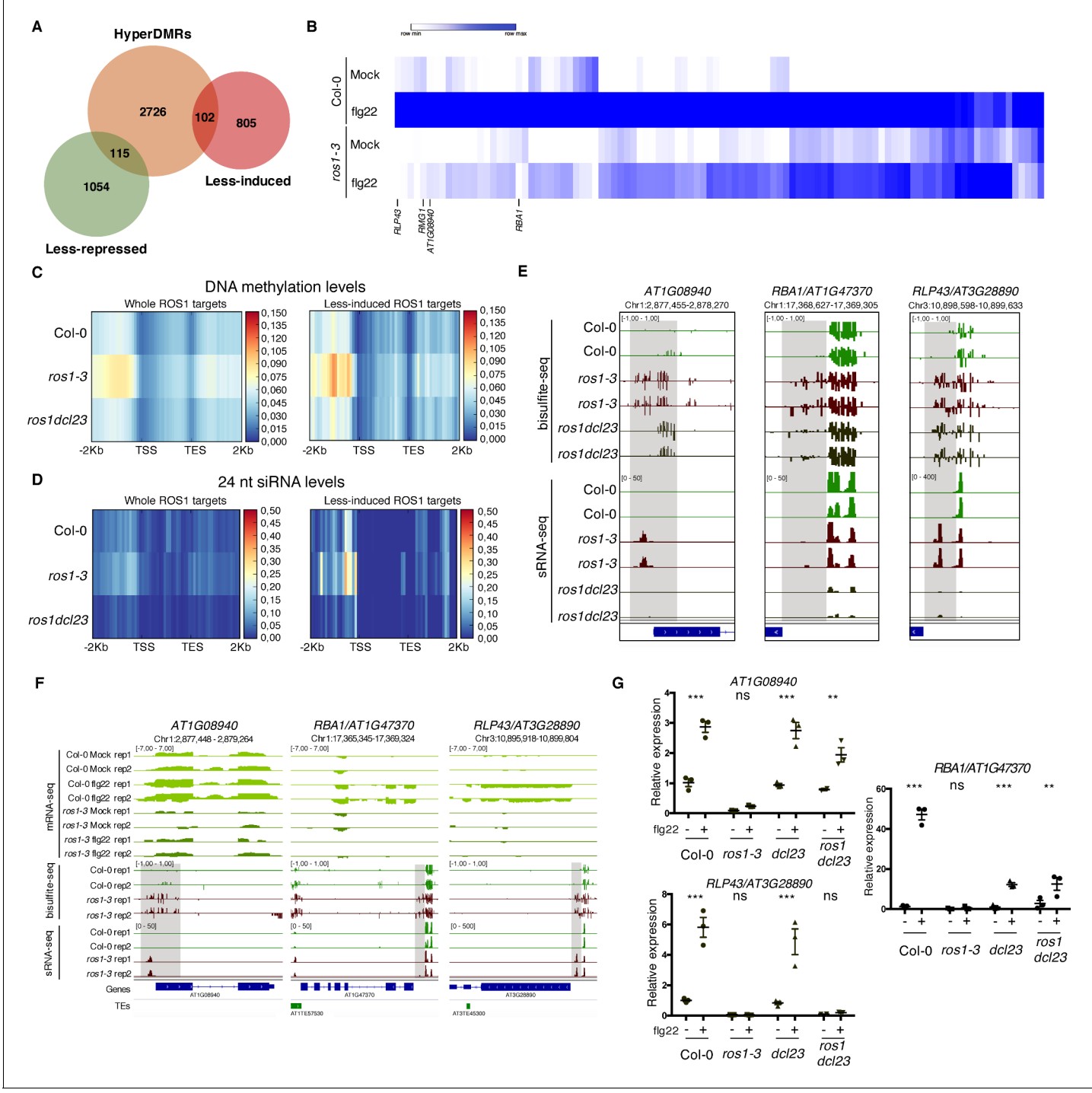

**Figure 3.** Genome-wide identification of flg22-responsive *ROS1* targets and characterisation of the role of RNA-directed DNA methylation (RdDM) in the methylation status of these genes in the absence of ROS1. (**A**) Proportion of flg22-responsive genes that are regulated by ROS1. One hundred and two flg22-responsive genes that are 'less-induced' and 115 flg22-responsive genes that are 'less-repressed' in *ros1-3*-elicited mutant exhibit hypermethylated DMRs (hyperDMRs). Venn diagram representing the overlap of genes presenting hyperDMRs in the *ros1* mutant (in orange) with genes presenting a compromised induction (in red) or repression (in green) in *ros1-3* compared to Col-0 treated with mock (water) or 1 µM of flg22 for 6 hr. (**B**) Heat map representing the relative expression of the 102 less-induced genes in Col-0 and *ros1-3* treated with mock (water) or 1 µM of flg22 for 6 hr. Merged data from two independent biological replicates are presented. (**C**) Increased global DNA methylation levels observed in untreated *ros1-3* compared to Col-0 at the whole set of genes exhibiting hyperDMRs (left panel) and at the 102 less-induced genes (right panel) are restored in the *ros1dcl23* triple mutant. Heatmap representing global DNA methylation levels within regions comprising 2 Kb upstream of the transcription start site (TSS), the gene body, and 2 Kb downstream of the transcription end site (TES) of the whole set of genes exhibiting hyperDMRs in *ros1-3 versus* Col-0

*Figure 3 continued on next page*

Figure 3 continued

(left panel) and of the 102 less-induced genes (right panel). These heatmaps were generated from BS-seq data sets obtained from 5-week-old rosette leaves of Col-0, *ros1-3*, and *ros1dcl23* mutants. (D) Increased 24 nt siRNA levels in *ros1-3* at the whole set of genes exhibiting hyperDMRs in *ros1-3 versus* Col-0 (left panel) and at the 102 less-induced genes (right panel) are restored in the *ros1dcl23* triple mutant. Heatmap representing 24-nt siRNA levels within regions comprising 2 Kb upstream of the TSS, the gene body, and 2 Kb downstream of the TES of the whole set of genes exhibiting hyperDMRs in *ros1-3 versus* Col-0 (left panel) and of the 102 less-induced genes (right panel). These heatmaps were generated from sRNA-seq datasets obtained from 5-week-old rosette leaves of untreated Col-0, *ros1-3* and *ros1dcl23* mutants. Average of the two replicates is represented. (E) Methylation levels are restored at *AT1G08940* and partially restored at *RBA1* in *ros1dcl23* whereas *RLP43* retains methylation levels similar to methylation levels observed in the single *ros1-3* mutant. IGV snapshots showing siRNA levels and methylation levels at the DMRs of *AT1G08940*, *RBA1*, and *RLP43* in Col-0, *ros1-3,* and *ros1dcl23*. (F) IGV snapshots depicting mRNA-seq data in Col-0 and *ros1-3* mutant in mock- and flg22-treated conditions as well as BS-seq and sRNA-seq of untreated Col-0 and *ros1-3* plants. (G) *AT1G08940* gene induction is restored and *RBA1* gene induction is partially restored whereas *RLP43* remains in a repressed state in *ros1dcl23*. RT-qPCR analyses from 5-week-old rosette leaves of Col-0, *ros1-3*, *dcl23*, and *ros1dcl23* treated with either mock (water) or 1 µM of flg22 for 6 hr. The mRNA levels are relative to the level of *UBQ* transcripts. Statistical significance of flg22 treatment on expression was assessed using a two-way ANOVA test and a Sidak's multiple comparisons test. Asterisks indicate statistical significance (*: $p<0.05$, **: $p<0.01$, ***: $p<0.001$, ns: not significant).

The online version of this article includes the following source data and figure supplement(s) for figure 3:

**Source data 1.** Original qRT-PCR data for *Figure 3G*.
**Source data 2.** Original transcriptomic data used for the heatmap presented in *Figure 3B*.
**Figure supplement 1.** RNA sequencing experiment in 5-week-old rosette leaves of Col-0 and *ros1-3* syringe-infiltrated with either mock or flg22 for 6 hr.
**Figure supplement 2.** Analysis of 23–24 nt siRNA levels for the groups of 102 less-induced genes and 2943 genes carrying hyperDMRs in the *ros1-3* mutant background.
**Figure supplement 3.** Gain of 21-nt siRNAs and P4RNAs at the *RLP43* promoter in the *ros1dcl23* background might contribute to the maintenance of DNA methylation levels in this triple mutant.

RNAs (25–29 nt), which exhibit a preference for having an A at their 5' ends and that overlap with siRNAs produced from the hyperDMR, as observed for typical P4RNA species (*Figure 3—figure supplement 3*; *Blevins et al., 2015*; *Zhai et al., 2015*; *Yang et al., 2016*). Because PolIV-dependent 21 nt siRNAs and P4RNAs are competent for RdDM (*Yang et al., 2016*; *Panda et al., 2020*), the above RNA entities might maintain RdDM at *RLP43* promoter in the *ros1dcl23* mutants, although RNA-independent mechanism(s) might additionally contribute to this process. Collectively, these data indicate that the flg22-triggered induction of these three genes is facilitated by ROS1, which prevents hypermethylation at discrete regions in their promoters. They also indicate that their promoter hypermethylation detected in *ros1-3* mutants can be fully or partially altered upon concomitant removal of DCL2 and DCL3, as found at the phosphoglycerate mutase and *RBA1* genes, respectively, or unchanged, as observed at the *RLP43* locus.

## WRKY TFs bind to a single W-box element embedded in the middle of the demethylated promoter region of *RLP43*, thereby ensuring a proper flg22-triggered induction of this gene

Given that the ROS1-directed demethylation of a subset of defence gene promoters was required for their flg22-triggered induction (*Figure 3*), we hypothesized that these demethylated promoter sequences might contain functional binding sites for PAMP-responsive TFs. To test this possibility, we selected 26 flg22-induced ROS1 targets exhibiting discrete and dense hypermethylated regions in *ros1-3* mutants, along with a strong impaired induction in *ros1-3*-elicited mutants, and subjected their hyperDMR sequences to Genome Association Tester (GAT) (*Figure 3—figure supplement 1*; *Heger et al., 2013*). This approach interrogates DNA affinity purification sequencing (DAP-seq) data sets of whole-genome Arabidopsis TF factor binding sites and assesses whether any DAP-seq peak would overlap more with the input DNA sequence than expected by chance (*O'Malley et al., 2016*). By using this analysis, we found *in vitro* DNA binding of WRKY TFs at 14 out of 26 promoter-derived sequences tested (*Figure 4A*, *Figure 4—figure supplement 1*). By contrast, only 2 out of 28 less-repressed stringent targets displayed WRKY TFs enrichment at their demethylated promoter regions (*Figure 4—figure supplement 2*). Among the 14 less-induced candidates, we retrieved *RMG1*, *RBA1*, and *RLP43*, which exhibit *in vitro* DNA binding of several WRKYs in their promoter regions subjected to ROS1-directed demethylation (*Figure 4A and B*, *Figure 4—figure supplement 1*, *Figure 4—figure supplement 3*, *Figure 4—figure supplement 4*). Five out of these 14 candidate

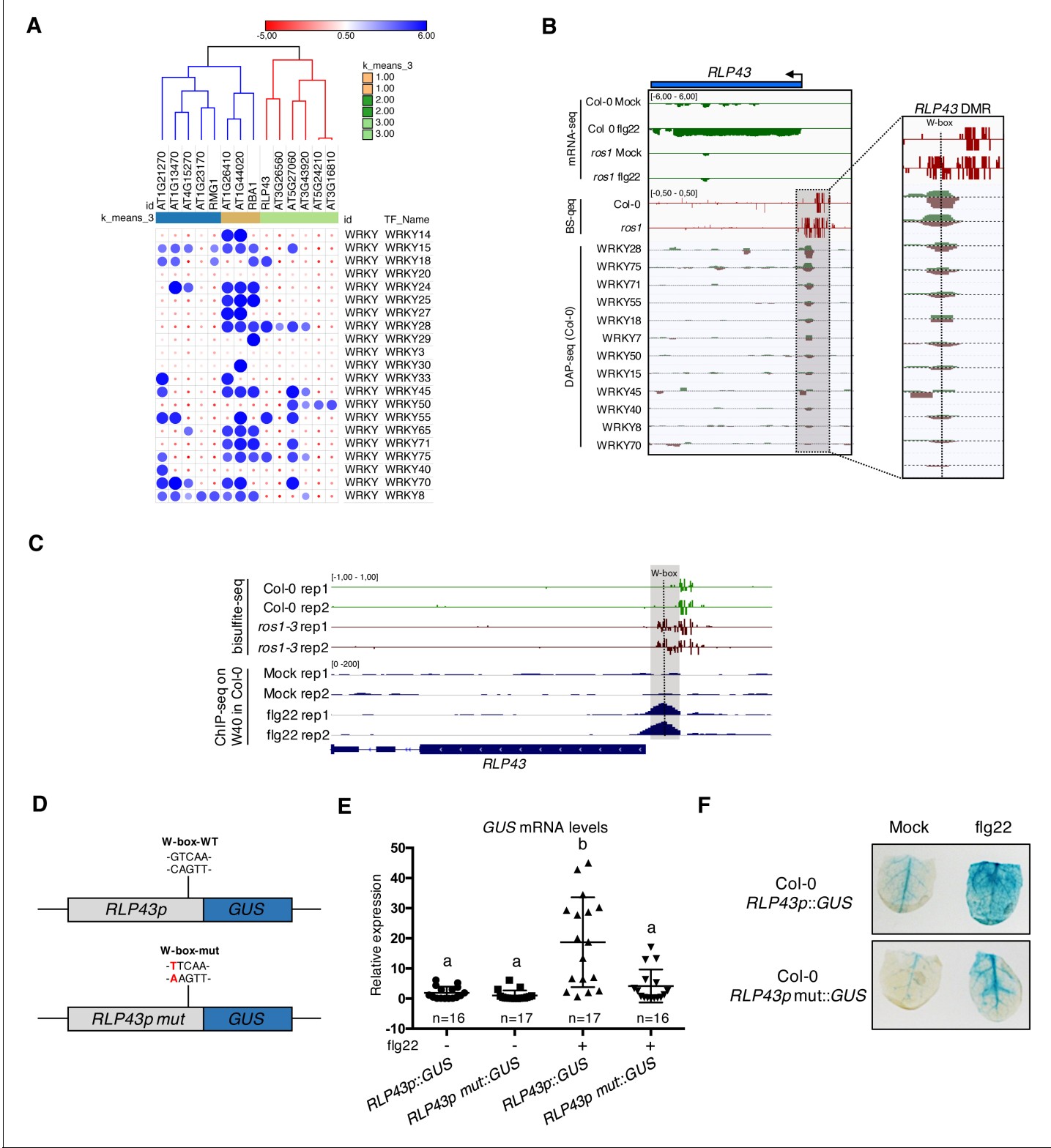

**Figure 4.** Several WRKY transcription factors bind to the demethylated region of the *RLP43* promoter, which contains a functional and flg22-responsive W-box cis-element. (**A**) A subgroup of the ROS1 targets exhibits an over-representation of WRKY DNA binding at the promoter regions corresponding to the hyperDMRs that were retrieved in *ros1-3*. GAT analysis performed on publicly available DNA affinity purification sequencing (DAP-seq) data (*O'Malley et al., 2016*) identified WRKY TFs with significant enrichment in the regions corresponding to the hyperDMRs observed in *ros1-3* for the 14 stringent ROS1 primary targets. (**B**) Several WRKY TFs specifically bind to the region that is demethylated by ROS1 in the *RLP43* promoter. Snapshots
*Figure 4 continued on next page*

*Figure 4 continued*

representing, from top to bottom, mRNA-seq (Rep1), BS-seq (Rep1), and DAP-seq data (*O'Malley et al., 2016*) at the *RLP43* locus. To better appreciate the overlap between the promoter region of *RLP43* subjected to ROS1-directed demethylation and the region where WRKY TFs bind to DNA, a zoom in is depicted at the level of the hyperDMR (box on the right panel) and the position of the W-box is highlighted by the vertical dashed line. (C) A hemagglutinin (HA) epitope-tagged version of WRKY40 binds *in vivo* to the ROS1-targeted region of the *RLP43* promoter in a flg22-dependent manner. Snapshots depicting bisulfite sequencing data from 5-week-old rosette leaves of Col-0 and *ros1-3* mutant (Rep1) and ChIP-seq data performed on seedlings of *wrky40* mutants (SLAT collection of dSpm insertion line; *Shen et al., 2007*) complemented with WRKY40-HA treated with either mock (medium without flg22) or flg22 (medium supplemented with flg22) for 2 hr, at *RLP43* (*Birkenbihl et al., 2017*). (D) Scheme representing the *RLP43* transcriptional GUS fusion constructs containing the WT W-box sequence (*RLP43::GUS*) and the mutated W-box sequence (*RLP43p mut::GUS*). (E) The W-box *cis*-element at the *RLP43* promoter is functional and responsive to flg22. RT-qPCR analyses were performed to monitor the *GUS* mRNA levels in primary T1 transformants expressing either the *RLP43p::GUS* or *RLP43p mut::GUS* transgenes. For each individual, two leaves were syringe-infiltrated with mock (water) and two other leaves were treated the same way with flg22 at a concentration of 1 μM for 6 hr. The *GUS* mRNA levels are relative to the level of *UBQ* transcripts. Statistical significance was assessed using a one-way ANOVA test and a Tukey's multiple comparisons test. (F) The flg22-induced GUS activity is impaired in *RLP43p mut::GUS*-elicited plants. GUS-staining of 5-week-old rosette leaves of *RLP43p::GUS* or *RLP43pmut::GUS* primary transformants that were syringe-infiltrated with either mock or 1 μM of flg22 for 24 hr. Representative pictures are shown.

The online version of this article includes the following source data and figure supplement(s) for figure 4:

**Source data 1.** Original qRT-PCR data of GUS transcript quantification for *Figure 4E*.
**Figure supplement 1.** GAT analysis of all the transcription factor classes binding to the 26 stringent less-induced ROS1 target promoters.
**Figure supplement 2.** GAT analysis of all the transcription factor classes binding to the 28 stringent less-repressed ROS1 target promoters.
**Figure supplement 3.** WRKY DNA binding peaks are present at the *RMG1* promoter region that is subjected to ROS1-directed demethylation.
**Figure supplement 4.** Position of WRKY DNA binding peaks at the 11 flg22-induced ROS1 targets.
**Figure supplement 5.** Flg22-triggered chromatin association of WRKY18-HA and WRKY40-HA at the promoters of five less-induced ROS1 targets and at the promoter of the positive control *bZIP60*.

promoters also exhibit *in vivo* binding of the AtWRKY18 and AtWRKY40 based on chromatin immunoprecipitation sequencing (ChIP-seq) data sets generated in flg22-challenged transgenic seedlings (*Figure 4—figure supplement 5*; *Birkenbihl et al., 2017*). Importantly, the chromatin association of these PAMP-responsive WRKYs was specifically detected during flg22 treatment, indicating that PTI signalling is required for this chromatin-based regulatory process (*Figure 4—figure supplement 5*). For example, the *in vivo* binding of AtWRKY40 at the *RLP43* promoter specifically occurs during flg22 elicitation and at the demethylated promoter region, which contains a single W-box *cis*-regulatory element (*Figure 4B and C*; *Birkenbihl et al., 2017*). This observation prompted us to characterize this W-box in the context of PTI. For this end, we generated transcriptional reporters by fusing either the WT or the W-box point mutant upstream sequences of *RLP43* with the β-glucuronidase *GUS* reporter gene (*Figure 4D*). We found a strong reduction in GUS staining and mRNA levels in flg22-challenged leaves of primary transformants expressing the W-box mutant- *versus* the WT-versions of the *RLP43* transcriptional reporters (*Figure 4E and F*). These results imply that this W-box *cis*-element is functional and essential for the flg22-triggered transcriptional activation of *RLP43*. Nevertheless, the remaining GUS staining detected in the leaf vasculature of flg22-treated plants expressing the W-box mutant transgene suggests that other TFs must additionally contribute to the flg22-triggered transcriptional activation of *RLP43* in these tissues (*Figure 4F*). Collectively, these data indicate that the W-box *cis*-element located in the demethylated region of the *RLP43* promoter is the site for WRKY TF binding. They also suggest that the flg22-triggered recruitment of WRKYs at the *RLP43* promoter is critical for the transcriptional activation of this gene during PTI.

## ROS1-directed removal of DNA methylation at the *RLP43* promoter is necessary for DNA binding of AtWRKY18 and AtWRKY40

DNA methylation has previously been shown to inhibit, in some instances, TF-DNA binding (*Watt and Molloy, 1988*; *Iguchi-Ariga and Schaffner, 1989*; *Tate and Bird, 1993*; *O'Malley et al., 2016*). For example, a DAP-seq study performed with Col-0 genomic DNA revealed that ~75% of Arabidopsis TFs are sensitive to DNA methylation (*O'Malley et al., 2016*). This was notably the case of several WRKY TFs used in their analysis (*Figure 5A*; *O'Malley et al., 2016*). This observation, along with the fact that the *ros1-3* mutation alters the flg22 induction of *RLP43* (*Figures 3* and *4*), suggests that the hypermethylation at its promoter observed in *ros1-3* mutants might interfere with WRKY TF-DNA binding. To test this hypothesis, we have used a DAP approach coupled with a

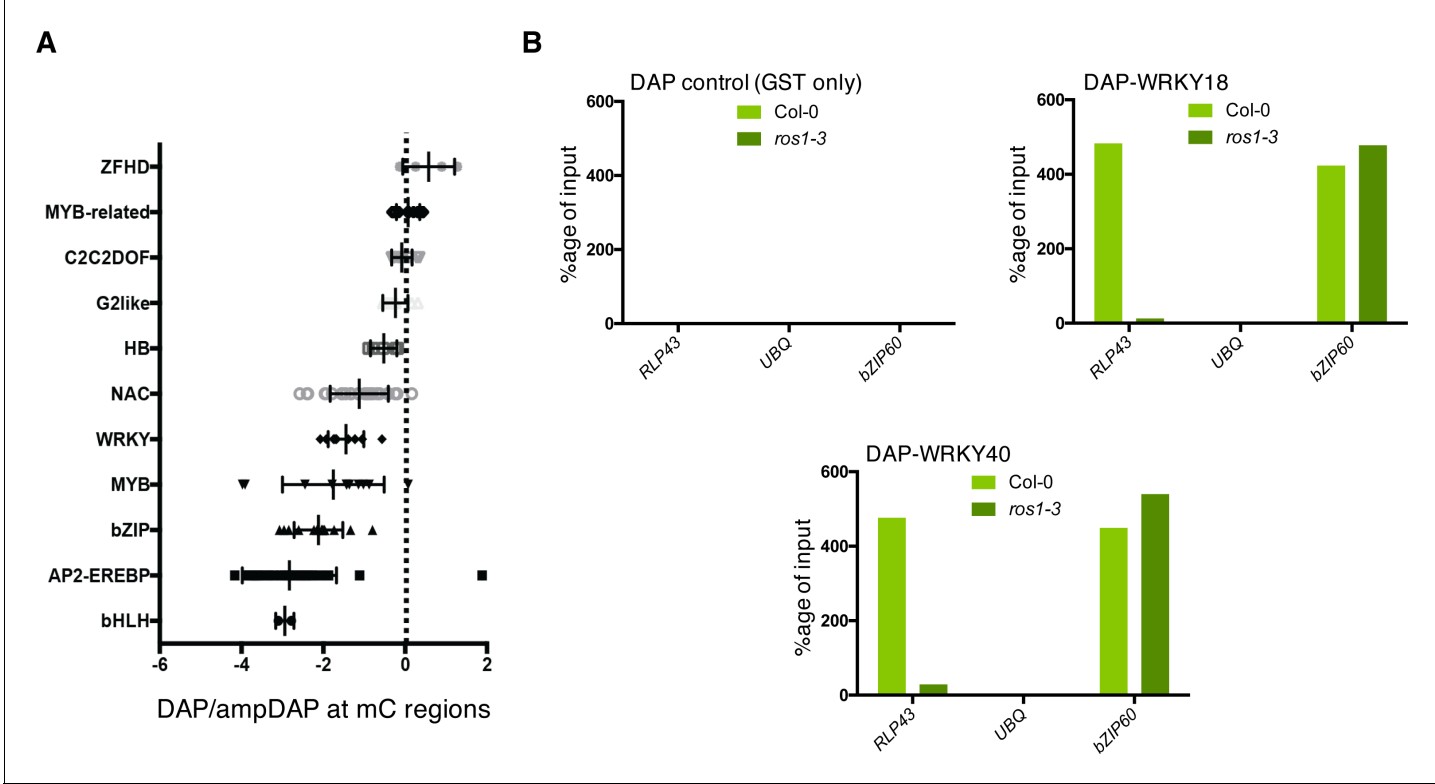

**Figure 5.** ROS1-directed demethylation is crucial for the DNA binding of WRKY18 and WRKY40 at the *RLP43* promoter. (**A**) DAP-seq data from *O'Malley et al., 2016* showing that WRKY family members are generally sensitive to DNA methylation. Graph representing ratio of binding capacity in DAP *versus* ampDAP data at regions methylated in Col-0 and for the different transcription factor family members used in this study. (**B**) The ability of WRKY18 and WRKY40 to bind DNA corresponding to the demethylated region of the *RLP43* promoter is abolished in the *ros1-3* mutant background. DAP-qPCR analysis at the *RLP43* promoter upon pull-down of Col-0 or *ros1-3* genomic DNA by GST (negative control), WRKY18-GST or WRKY40-GST. UBQ and *bZIP60* served as negative and positive controls, respectively.

The online version of this article includes the following source data and figure supplement(s) for figure 5:

**Source data 1.** Original DAP-qPCR data for *Figure 5B*.

**Figure supplement 1.** ROS1-directed demethylation is crucial for the binding of WRKY18 and WRKY40 at the *RLP43* promoter.

**Figure supplement 1—source data 1.** Original DAP-qPCR data for *Figure 5—figure supplement 1*.

quantitative PCR (DAP-qPCR) analysis. More specifically, we have expressed in bacteria Glutathione S-transferase (GST)-tagged AtWRKY18 and AtWRKY40, two well-characterized Arabidopsis WRKYs that are PAMP-responsive (*Birkenbihl et al., 2017*). Affinity-purified WRKYs were subsequently incubated with sheared genomic DNA from either Col-0 or *ros1-3* mutants and we further monitored WRKY DNA binding by qPCR analysis at the *RLP43* promoter. The rationale for using *RLP43* promoter for this specific analysis was based on (1) previous DAP-seq and ChIP-seq data showing robust binding of different WRKY TFs at the ROS1-directed demethylated *RLP43* promoter (*Figure 4A–C*, *Figure 4—figure supplement 1*), (2) the presence of a single W-box *cis*-element located right in the middle of this genomic region (*Figure 4B and C*), and (3) the central role of this W-box in the flg22-responsiveness of the *RLP43* promoter (*Figure 4D–F*). As a positive control for WRKY binding, we also amplified a promoter region of *bZIP60*, which is unmethylated in *ros1-3* mutants and bound by different PAMP-responsive WRKY TFs (*Figure 2—figure supplement 2*, *Figure 4—figure supplement 5*; *Birkenbihl et al., 2017*). Using this DAP-qPCR approach, we found that both WRKY TFs exhibit strong and comparable enrichment at the *RLP43* and *bZIP60* promoters in the presence of Col-0 genomic DNA (*Figure 5B*, *Figure 5—figure supplement 1*), while no binding was found with GST alone (*Figure 5B*, *Figure 5—figure supplement 1*). By contrast, the DNA binding of these Arabidopsis WRKY TFs was almost fully abolished at the *RLP43* promoter with genomic DNA from *ros1-3* mutants, while it remained unaltered at the *bZIP60* promoter in the same conditions (*Figure 5B*,

*Figure 5—figure supplement 1*). Collectively, these data provide evidence that the hypermethylation observed in *ros1-3* mutants at the *RLP43* promoter directly repels DNA binding of WRKY TFs.

## ROS1-directed demethylation of *RMG1* and *RLP43* promoter-regulatory regions is causal for both flg22-triggered gene inducibility as well as basal resistance against *Pto* DC3000

The fact that the demethylation of a discrete *RLP43* promoter region, carrying a functional W-box *cis*-element, is required for WRKY TF DNA binding, suggests that the ROS1-directed suppression of promoter hypermethylation must be causal for flg22-triggered gene inducibility. To test this hypothesis, we generated Arabidopsis transgenic lines expressing a chimeric inverted repeat transgene bearing sequence identity to the *RMG1* and *RLP43* promoter regions that are subjected to ROS1-dependent demethylation (*Figure 6A*). By using this approach, we aimed to assess whether artificial siRNAs in Col-0 would override the dominance of ROS1 over RdDM, thereby triggering hypermethylation at these promoter regions, which carry W-box *cis*-elements and DNA binding site of WRKY TFs (*Figure 4*, *Figure 4—figure supplement 3*). Molecular characterization by northern blot and McrBC-qPCR analyses, respectively, revealed levels of siRNA accumulation and of methylation at these promoter regions that were comparable to the ones detected in *ros1-3* mutants (*Figure 6B*, *Figure 6—figure supplement 1*), thereby validating our experimental strategy. We next challenged two independent IR-*RMG1p/RLP43p* transgenic lines with flg22 and monitored the levels of *RMG1* and *RLP43* mRNAs at 6 hr post-treatment. Significantly, we found that the induction of these genes was strongly altered in the two IR-*RMG1p/RLP43p* independent transgenic lines, almost to the same extent as in the *ros1*-elicited mutant background (*Figure 6C*, *Figure 6—figure supplement 1C*). By contrast, the flg22-triggered induction of *RBA1* and *FRK1* was comparable in Col-0 and IR-*RMG1p/RLP43p* transgenic lines (*Figure 6D*, *Figure 6—figure supplement 1B*), supporting a specific transcriptional silencing effect towards *RMG1* and *RLP43*. In addition, we found that IR-*RMG1p/RLP43p* lines displayed enhanced *Pto* DC3000 titres compared to Col-0-infected plants (*Figure 6E*), a phenotype resembling the one observed in *rmg1*-infected plants (*Figure 2C*). Collectively, these data demonstrate that hypermethylation of these *RMG1* and *RLP43* discrete promoter regions is causal for the compromised flg22 induction of these genes and for increased susceptibility towards *Pto* DC3000. We conclude that ROS1-directed demethylation of these promoter regulatory sequences is critical for both flg22-triggered gene inducibility and antibacterial resistance.

## Discussion

Active demethylation has emerged as a key regulatory mechanism of plant disease resistance. For examples, an enhanced susceptibility towards *Pto* DC3000 and *Hyaloperonospora arabidopsidis* was reported in *ros1* mutants (*Yu et al., 2013*; *López Sánchez et al., 2016*). A heightened disease susceptibility towards *Fusarium oxysporum* was also found in the Arabidopsis *ros1 dml2 dml3* (*rdd*) mutant, and this phenotype was exacerbated by knocking-down *DME* in this background (*Le et al., 2014*; *Schumann et al., 2019*). Although these data indicate that demethylases promote basal resistance against unrelated phytopathogens, very little is known about the mechanisms involved in this process. Here, we showed that Arabidopsis *ros1* mutants exhibited enhanced growth and vascular propagation of *Pto* DC3000, two phenotypes which were almost fully rescued in *ros1dcl23*-infected mutants (*Figure 1*). Therefore, ROS1 promotes basal resistance against *Pto* DC3000 mainly by antagonizing *DCL2* and/or *DCL3* functions.

We further found that the reprogramming of more than 2000 genes was altered in *ros1*-elicited mutants, supporting a major role for ROS1 in controlling PAMP-triggered gene expression changes. Among these genes, only ~10% exhibited hyperDMRs in *ros1* mutants and were thus potentially directly regulated by ROS1. This is congruent with a previous transcriptome study conducted in *ros1* mutants infected with *Hyaloperonospora arabidopsidis* (*López Sánchez et al., 2016*) and suggests that the majority of biotic stress-responsive genes are ROS1 secondary targets. This phenomenon is likely explained by the fact that ROS1 directly controls the expression of a subset of central regulatory components of immune responses, such as plant immune receptors, which have large impact on downstream signalling events.

In addition, we found that a similar proportion of ROS1 putative primary targets were up-regulated (102 genes) and down-regulated (115 genes) in response to flg22. These results suggest that

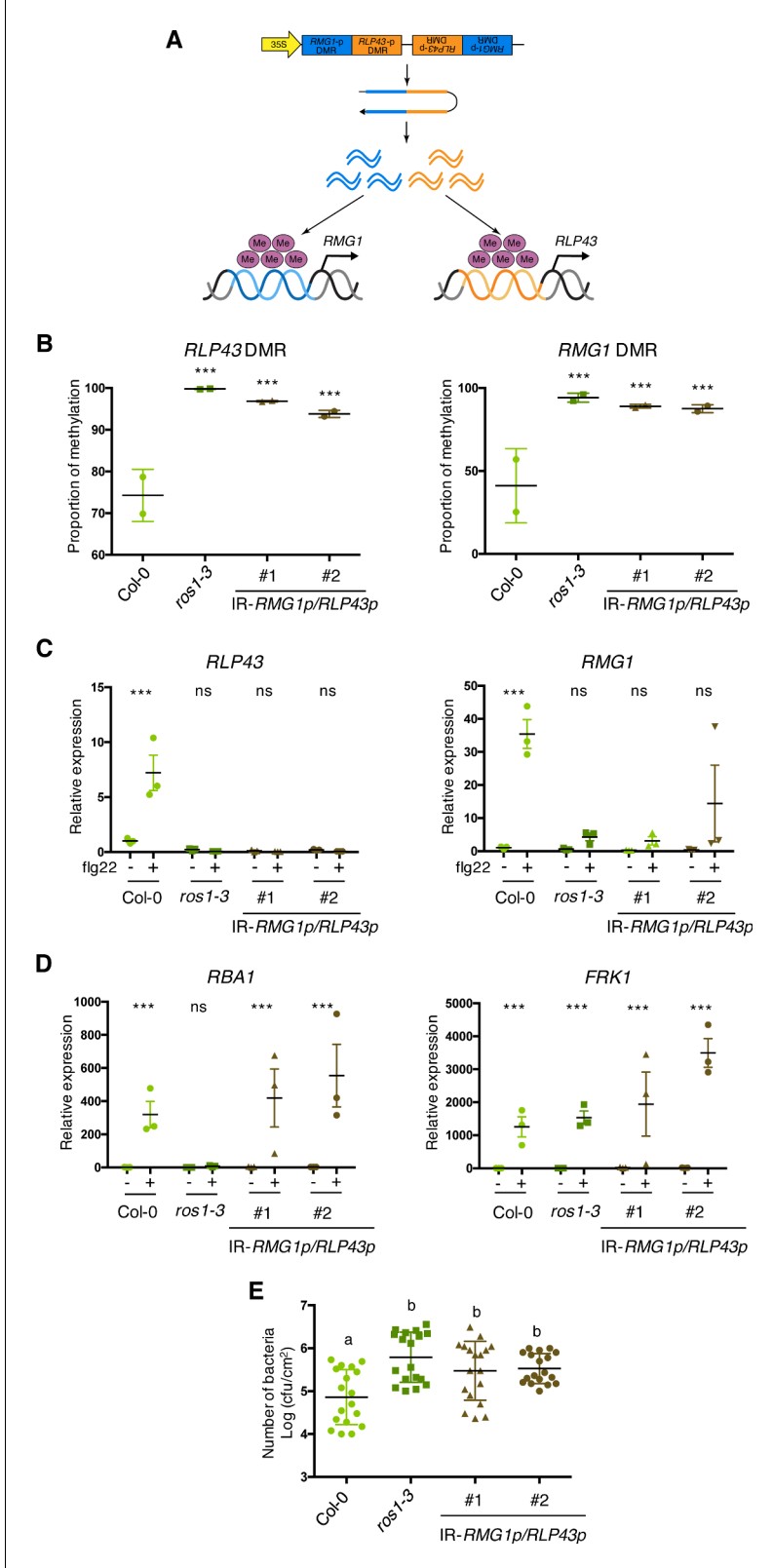

**Figure 6.** The artificial siRNA-directed targeting of remethylation at the *RMG1* and *RLP43* promoters impairs the flg22-triggered inducibility of these genes and enhances susceptibility towards *Pto* DC3000. (**A**) Scheme depicting the chimeric inverted repeat (IR) construct designed to simultaneously direct DNA remethylation at the *RMG1* and *RLP43* promoter regions. The IR-*RMG1p/RLP43p* contains the sequences corresponding exactly to the promoter sequence regions of *RMG1* (blue) and *RLP43* (orange) that are subjected to ROS1-directed demethylation in Col-0 and hypermethylated in

*Figure 6 continued on next page*

*Figure 6 continued*

*ros1* mutants. This inverted repeat transgene is driven by the constitutive 35S promoter, hence hypothesized to constitutively produce two populations of siRNA species designed to force remethylation of the *RMG1* and *RLP43* promoter regions that are normally demethylated by ROS1 in Col-0. (**B**) The *RMG1* and *RLP43* promoters exhibit hypermethylation in IR-*RMG1p/RLP43p* lines such as in *ros1-3* mutants. Genomic DNAs from Col-0, *ros1-3,* and two independent IR-*RMG1p/RLP43p* lines (two biological replicates per line) were digested using McrBC and further analysed by qPCR. Ratio between digested DNA and undigested DNA was quantified to assess the proportion of methylation. (**C**) The flg22-triggered induction of *RMG1* and *RLP43* is impaired in the two independent IR-*RMG1p/RLP43p* lines. Five-week old rosette leaves of Col-0, *ros1-3*, and two independent IR-*RMG1p/RLP43p* lines were syringe-infiltrated with either mock (water) or 1 µM of flg22 for 6 hr, and the mRNA levels of *RMG1* and *RLP43* were monitored by RT-qPCR analyses. The mRNA levels are relative to the level of *UBQ* transcripts. Statistical significance of flg22 treatment on expression was assessed using a two-way ANOVA test and a Sidak's multiple comparisons test. Asterisks indicate statistical significance (*: p<0.05, **: p<0.01, ***: p<0.001, ns: not significant). (**D**) The flg22-triggered induction of *RBA1* and *FRK1* is not affected in the two independent IR-*RMG1p/RLP43p* lines. Five-week old rosette leaves of Col-0, *ros1-3*, and two independent IR-*RMG1p/RLP43p* lines were syringe-infiltrated with either mock (water) or 1 µM of flg22 for 6 hr, and the mRNA levels of *RBA1* and *FRK1* were monitored by RT-qPCR analyses. The mRNA levels are relative to the level of *UBQ* transcripts. Statistical significance of flg22 treatment on expression was assessed using a two-way ANOVA test and a Sidak's multiple comparisons test. Asterisks indicate statistical significance (*: p<0.05, **: p<0.01, ***: p<0.001, ns: not significant). (**E**) IR-*RMG1p/RLP43p* lines exhibit increased *Pto* DC3000 titre. Five-week-old plants of Col-0, *ros1-3*, and two independent IR-*RMG1p/RLP43p* lines were dip-inoculated with *Pto* DC3000-GFP at $5 \times 10^7$ cfu ml$^{-1}$. Bacterial titres were monitored at 3 days post-infection (dpi). Each data point represents bacterial titre extracted from a single leaf. Three leaves out of four plants per line and from three independent experiments were considered for the comparative analysis. Statistical significance was assessed using a one-way ANOVA test and Tukey's multiple comparisons test.

The online version of this article includes the following source data and figure supplement(s) for figure 6:

**Source data 1.** Original McrBC-qPCR, qRT-PCR, and bacterial propagation data for *Figure 6B–E*.

**Figure supplement 1.** Artificial siRNA-directed remethylation of *RMG1* and *RLP43* promoters in the Col-0 background limits flg22-triggered induction of these genes.

**Figure supplement 1—source data 1.** Original qRT-PCR data for *Figure 6—figure supplement 1*.

ROS1 is equally capable of promoting the induction and repression of genes during flg22 elicitation. Some of these repressed targets might act as plant defence repressors or, alternatively, as modulators of plant developmental or physiological processes, possibly to redirect resources towards immunity and away from growth. It is noteworthy that the role of active demethylation in promoting repression of genes has previously been reported in the context of fruit ripening, whereby the tomato *ROS1* homolog *SlDML2* was found to repress hundreds of ripening-related genes (*Lang et al., 2017*). Further studies are needed to determine whether this feature of active demethylases could occur in the context of other biological processes.

We also performed a detailed analysis of the flg22-induced ROS1 putative primary targets. Most of these genes exhibited hypermethylation at their promoters, an effect that was almost abolished in *ros1dcl23* mutants (*Figure 3C and D*). These data suggest that ROS1 antagonizes RdDM at these promoters, presumably to facilitate their transcriptional activation during plant defence signalling. Consistent with this hypothesis, we found that the impaired induction of *RMG1*, *RBA1* and of the phosphoglycerate mutase genes observed in *ros1*-elicited plants was either fully or partially restored in *ros1dcl23*-elicited mutants (*Figures 2* and *3*). This was, however, not the case of *RLP43*, which remained fully silenced in *ros1dcl23*-elicited mutants, a phenotype that was associated with a moderate gain of 21 nt siRNAs at the *RLP43* hyperDMR in *ros1dcl23* mutants (*Figure 3*, *Figure 3—figure supplement 3*). This observation suggests that *DCL4* and/or *DCL1* might compensate for the lack of *DCL2* and *DCL3* in the *ros1* mutants to maintain RdDM at the *RLP43* promoter. This would be consistent with recent findings showing that PolIV-dependent 21nt siRNA species, which are produced by DCL4, can trigger RdDM (*Panda et al., 2020*). The longer RNAs detected at the *RLP43* hyperDMR in the *ros1dcl23* mutants (*Figure 3*, *Figure 3—figure supplement 3*), and which exhibited classical features of P4RNAs (*Figure 3—figure supplement 3*; *Blevins et al., 2015*; *Zhai et al., 2015*), might also contribute to this phenomenon. The latter hypothesis is supported by the fact that P4RNA species can efficiently direct DNA methylation at a large set of RdDM targets in the absence of DCL-dependent siRNAs (*Yang et al., 2016*). Alternatively, and/or additionally, some yet-unknown RNA-independent mechanism(s) might ensure the maintenance of DNA methylation at this *locus* in the absence of ROS1, DCL2, and DCL3 factors.

Interestingly, our work demonstrated that the dominance of ROS1 over RdDM at two immune-responsive gene promoters can be alleviated by artificially producing siRNAs against these genomic regions (*Figure 6*). In particular, we showed that the stable expression of an inverted repeat

transgene bearing sequence homologies to the ROS1 targeted regions of *RMG1* and *RLP43* promoters, resulted in their hypermethylation along with a compromised flg22-triggered induction of these genes (*Figure 6*). Furthermore, we found that these transgenic lines exhibited an enhanced susceptibility towards *Pto* DC3000, indicating that the silencing of these genes is sufficient to dampen basal resistance (*Figure 6*). We are anticipating that this approach will be extensively used in the future to assess the regulatory function of demethylases over any target(s) of interest. This strategy could also be exploited to transmit, and possibly maintain, epialleles at specific demethylase target(s), and perhaps even when the transgene is segregated away (if the intrinsic features of the targeted sequences are prone to form stable epialleles) (*Catoni et al., 2017*; *Li et al., 2020*). Such an epi-editing strategy would be particularly relevant in crops to prevent the unwanted expression of demethylase target(s) having negative effects on desirable trait(s).

Although the inactivation of demethylases is known to dampen resistance towards unrelated phytopathogens (*Yu et al., 2013*; *López Sánchez et al., 2016*; *Le et al., 2014*; *Schumann et al., 2019*), the functional relevance of individual demethylase targets in basal resistance remains ill-defined. Here, we have conducted an in-depth characterization of the ROS1 target *RMG1,* an orphan *TNL* gene containing two remnant RC/Helitron TEs in its promoter. The *AtREP4* distal repeat was found highly methylated in all cytosine sequence contexts, a methylation pattern that was required to maintain a low basal expression of this gene (possibly to prevent trade-off effects caused by its overexpression) (*Yu et al., 2013*; *Deleris et al., 2016*). The *AtREP11* proximal repeat was, conversely, unmethylated in Col-0 plants and gained high levels of methylation in *ros1* mutants, particularly at the 3' boundary of the *AtREP11* repeat, which contains W-box *cis*-elements and displays DNA binding of WRKY TFs (*Figure 2*, *Figure 4—figure supplement 4*; *Yu et al., 2013*). Significantly, both the accumulation of siRNAs and the hypermethylation observed in *ros1* mutant at this promoter region were found abolished in the *ros1dcl23* mutant background (*Figure 2*). Furthermore, a full restoration of flg22-triggered inducibility of *RMG1* was observed in *ros1dcl23*-elicited mutants, indicating that siRNAs limit the induction of this gene in the *ros1*-elicited mutant background (*Figure 2*). Importantly, we found that mutations in *RMG1* enhanced disease susceptibility towards *Pto* DC3000, highlighting a major role of this *TNL* in basal resistance towards this bacterium (*Figure 2*). These data support previous findings showing that *NLRs* are not solely required for race-specific resistance but can also contribute to basal resistance, possibly by orchestrating PTI signalling and/or by mounting a weak ETI response through the recognition of virulence factors (*Boccara et al., 2014*; *Canto-Pastor et al., 2019*; *Roth et al., 2017*). Although *RMG1* is a functional ROS1 target, the concomitant silencing of several other defence genes in *ros1* must additionally account for the increased disease susceptibility phenotypes observed in this mutant background.

DNA methylation has been shown to block DNA binding of some TFs *in vitro* (*Watt and Molloy, 1988*; *Iguchi-Ariga and Schaffner, 1989*; *Tate and Bird, 1993*; *O'Malley et al., 2016*). However, recent mechanistic studies at the promoters and/or enhancers of mammalian genes have highlighted a more complex picture, whereby TFs are either sensitive to DNA methylation or, alternatively, bind to methylated regions and in turn trigger their demethylation (*Domcke et al., 2015*; *Schübeler, 2015*; *Yin et al., 2017*). In human cells, earlier studies suggested a role for active demethylation in modulating the chromatin accessibility of TFs in the context of bacterial elicitation or infection (*Pacis et al., 2015*). For example, a large set of genomic regions were found demethylated in human dendritic cells infected with *Mycobacterium tuberculosis*, leading to chromatin relaxation at enhancers carrying stress-responsive *cis*-elements (*Pacis et al., 2015*). Nevertheless, a recent study suggests that demethylation at these immune-responsive enhancers is unlikely required for the chromatin-based recruitment of TFs in these conditions, but is more probably occurring as a consequence of TFs binding at these genomic regions (*Pacis et al., 2019*). However, the exact role that demethylation could play in the DNA/chromatin accessibility for TFs remains ambiguous, because this work has not been conducted in human cells lacking individual demethylases. Here, we have exploited the genetically tractable model organism *Arabidopsis thaliana* to investigate this question at immune-responsive gene promoters. Interestingly, we found that the flg22-induced ROS1 primary targets exhibit an over-representation of *in vitro* WRKY TFs binding at discrete demethylated promoter regions (*Figure 4*, *Figure 4—figure supplement 1*). Some of these W-box elements are likely functional because *in vivo* recruitment of WRKYs was additionally observed at a subset of promoter regions in flg22-treated seedlings (*Figure 4*, *Figure 4—figure supplement 5*; *Birkenbihl et al., 2017*). By using a transcriptional fusion reporter approach, we also provided evidence that a single

W-box element, centred in the *RLP43* promoter region that is demethylated by ROS1, was essential for flg22-triggered induction of this gene (*Figure 4*). Furthermore, we demonstrated that ROS1-directed demethylation of this *RLP43* promoter region facilitated DNA binding of the PAMP-responsive AtWRKY40 and AtWRKY18 (*Figure 5*, *Figure 5—figure supplement 1*; *Birkenbihl et al., 2017*). Based on these findings, we propose that ROS1 *cis*-regulates a subset of defence genes by ensuring binding of WRKY TFs at promoter-regulatory regions, thereby allowing their rapid and pervasive transcription during PTI. Consistent with this *cis*-effect model, we observed a compromised flg22-triggered inducibility of *RLP43* and *RMG1* by forcing methylation at discrete promoter-regulatory regions, which are normally demethylated by ROS1, and that carry W-box elements (*Figure 4*, *Figure 6*, *Figure 6—figure supplement 1*). ROS1-directed demethylation of defence gene promoters therefore likely provides a chromatin environment that is permissive for WRKY TFs binding, rendering these genes poised for transcriptional activation during plant defence signalling. It is noteworthy that the proposed mechanism might also hold true for other TF families, which exhibit *in vitro* binding at the demethylated promoter regions of flg22-induced ROS1 targets (*Figure 4—figure supplement 1*). Furthermore, this phenomenon might contribute to the down-regulation of genes during PTI, by facilitating the recruitment of repressive TFs at their promoters. Consistent with this hypothesis, different classes of TFs were found to preferentially bind to demethylated promoter regions of a subset of flg22-repressed ROS1 targets (*Figure 4—figure supplement 2*).

Plants and animals have evolved sophisticated epigenetic reprogramming prior and after fertilization. In mammals, two waves of global demethylation and remethylation occur during germ cell development and embryogenesis, which tightly control transgenerational epigenetic inheritance (*Heard and Martienssen, 2014*). In plants, epigenetic reprogramming during sexual reproduction is less robust than in mammals, rendering epiallele transmission more possible through generations. This is supported by the description of multiple artificial and natural epialleles, recovered from various plant species (*Weigel and Colot, 2012*; *Heard and Martienssen, 2014*). In Arabidopsis, the dynamics and mechanisms of epigenetic reprogramming have been extensively characterized during the last decade. While high levels of CG methylation are observed in male meiocytes, microspores, and sperm cells, a substantial erasure of CHH methylation occurs in the male germline (*Calarco et al., 2012*; *Ibarra et al., 2012*; *Walker et al., 2018*), which is regained during embryogenesis through RdDM (*Bouyer et al., 2017*; *Kawakatsu et al., 2017*). The latter phenomenon is accompanied by a restoration of active demethylation, which orchestrates removal of DNA methylation at incoming TEs/repeats and imprinted genes that display CG hypermethylation in the sperm cells, as a result of ROS1/DMLs repression in these cell types. Similarly, ROS1/DMLs might restore demethylation at defence gene promoters in the embryo, notably to ensure the reestablishment of poised W-box elements in the offspring (*Deleris et al., 2016*). Conversely, enhanced accumulation of endogenous siRNAs at these regulatory sequences is predicted to overcome demethylase reprogramming, as observed upon artificial production of siRNAs against *RMG1* and *RLP43* promoter regions carrying W-box *cis*-elements (*Figure 3*, *Figure 6*, *Figure 6—figure supplement 1*), and would likewise favour epiallele formation. Investigating whether such hypothetical mechanism could occur in nature and contribute to the emergence of immune-responsive expression variants will be essential to unravel the mechanisms by which environmental constraints drive the selection of new phenotypes during plant evolution and adaptation.

## Materials and methods

### Key resources table

| Reagent type (species) or resource | Designation | Source or reference | Identifiers | Additional information |
|---|---|---|---|---|
| Gene (*Arabidopsis thaliana*) | *ROS1* | Arabidopsis.org | AT2G36490 | |
| Gene (*Arabidopsis thaliana*) | *DCL2* | Arabidopsis.org | AT3G03300 | |
| Gene (*Arabidopsis thaliana*) | *DCL3* | Arabidopsis.org | AT3G43920 | |

*Continued on next page*

*Continued*

| Reagent type (species) or resource | Designation | Source or reference | Identifiers | Additional information |
|---|---|---|---|---|
| Gene (*Arabidopsis thaliana*) | *RMG1* | Arabidopsis.org | *AT4G11170* | |
| Gene (*Arabidopsis thaliana*) | *RLP43* | Arabidopsis.org | *AT3G28890* | |
| Strain, strain background (*Pseudomonas syringae* pv. *tomato*) | *Pto* DC3000-GFP | Gift from Pr. Sheng Yang He (Duke University, US). | | |
| Genetic reagent (*Arabidopsis thaliana*) | *ros1-3* | *Penterman et al., 2007* | | |
| Genetic reagent (*Arabidopsis thaliana*) | *ros1-4* | Nottingham Arabidopsis stock center (NASC) | N682295 | |
| Genetic reagent (*Arabidopsis thaliana*) | *dcl2-1 dcl3-1* | *Xie et al., 2004* | | |
| Genetic reagent (*Arabidopsis thaliana*) | *ros1-3 dcl2-1 dcl3-1* | This study | | |
| Genetic reagent (*Arabidopsis thaliana*) | *rmg1-1* | Nottingham Arabidopsis stock center (NASC) | N678063 | |
| Genetic reagent (*Arabidopsis thaliana*) | *rmg1-2* | Nottingham Arabidopsis stock center (NASC) | N674117 | |
| Commercial assay or kit | MagneGST Pull-Down System | Promega | V8870 | |
| Commercial assay or kit | McrBC | New England Biolab (NEB) | M0272 | |

## Plant material and growth conditions

Arabidopsis plants were grown in short day condition (8 hr light at 22.5°C, 16 hr dark at 19.5°C) and all experiments were performed on 5-week-old rosette leaves.

## Mutant lines

Besides WT Col-0, the single mutants *ros1-3* (*Penterman et al., 2007*), *ros1-4* (SALK_045303), *rmg1-1* (SALK_007034) and *rmg1-2* (SALK_023944), the double *dcl2-1 dcl3-1* (*dcl23*) (*Xie et al., 2004*), and the triple *ros1-3 dcl2-1 dcl3-1* (*ros1dcl23*) mutants were used in this study.

## Plasmids and constructs

### The WRKY-GST constructs

The pGEX-2TM-WRKY18 and pGEX-2TM-WRKY40 plasmids allow bacterial expression of the N-ter GST-tagged version of full-length WRKY18 and WRKY40 (gift from Pr. Imre Somssich, MPI, Cologne, Germany). Full-length cDNAs of each WRKYs were amplified and cloned into Gateway compatible entry vectors and subsequently recombined into the pGEX-2TM-GW destination vector.

### The IR-*RMG1p/RLP43p* inverted repeat construct

The IR-*RMG1p/RLP43p* construct is a chimeric inverted repeat construct composed in each arm of the 180 bp and 137 bp sequences that are demethylated by ROS1 at the *RMG1* and *RLP43* promoters, respectively (with the intron of the petunia chalcone synthase gene *CHSA* in between). The IR-*RMG1p/RLP43p* was synthesized and inserted by restriction enzyme digestion into the modified pDON221-P5-P2 vector by GenScript. A double recombination of this vector together with the pDON221-P1-P5r plasmid carrying the Cauliflower Mosaic Virus (CaMV) 35S promoter sequence in the pB7WG Gateway destination vector was performed to obtain the destination plasmid containing the 35Sp::IR-*RMG1p/RLP43p* construct. The plasmid was further introduced into the *Agrobacterium*

*tumefaciens* C58C1 strain, which was then used for Agrobacterium-mediated plant transformation (*Clough and Bent, 1998*).

### *RLP43p*::*GUS* and *RLP43pmut*::*GUS*

The 1.5 Kb sequence located upstream of the start codon of *RLP43* was cloned in the pENTR/D-TOPO vector and recombined in the pBGWFS7 binary destination vector, containing the *GUS* reporter gene. Site-directed mutagenesis was performed with primers GAACGCCTCGTAGG TTCAAGCTGTGTTGGAAT and ATTCCAACACAGCTTGAACCTACGAGGCGTTC to mutate the W-box from GGTCAA to GTTCAA. Both *RLP43p*::*GUS* and *RLP43pmut*::*GUS* constructs were transformed in the Col-0 accession by Agrobacterium-mediated method (*Clough and Bent, 1998*). Primary transformants were selected with Basta herbicide (10 µg/mL). GUS staining as well as RT-qPCR analysis on *GUS* transcripts were performed on at least 16 individual primary transformants for WT and mutated constructs. The primary transformants were infiltrated with either mock (water) or 1 µM of flg22 during 6 hr for the qRT-PCR and 24 hr for the GUS staining.

## Bacterial secondary leaf vein inoculation and quantification of the bacterial spreading phenotypes

Bacterial propagation in the secondary veins was assessed as described previously (*Yu et al., 2013*). About 12 leaves from three plants per condition were inoculated with a toothpick that was previously dipped in an inoculum of GFP-tagged *Pto* DC3000 at a concentration of $5 \times 10^6$ cfu/ml ($OD_{600}$ of 0.2 corresponds to $10^8$ cfu/ml). Inoculation was done on six secondary veins per leaf and two sites of inoculation per vein. The number of bacterial spreading events from the wound inoculation sites was quantified after 3 days under UV light using a macrozoom (Olympus MVX10). When the bacteria propagated away from any of the 12 inoculation sites, it was indexed as propagation with a possibility of maximum 18 propagations per leaf. The values from three independent experiments were considered for the comparative analysis. Statistical significance was assessed using a one-way ANOVA test and Tukey's multiple comparisons test.

## Bacterial growth assays

Four plants per condition were dip-inoculated using *Pto* DC3000 at $5 \times 10^7$ cfu/ml supplemented with 0.02% Silwet L-77, and immediately placed in chambers with high humidity. Bacterial growth was determined 3 days post-infection. For the quantification, infected leaves were harvested, washed for 1 min in 70% (v/v) EtOH, and 1 min in water. Leaf discs with a diameter of 4 mm were excised, grinded, and homogenized in 100 µl of water. Each data point consists of four-leaf discs. Fifteen microliters of each homogenate were then plated undiluted and at different dilutions. Bacterial growth was determined after 36 hr of incubation at 28°C by colony counting. The values from three independent experiments were considered for the comparative analysis. Statistical significance was assessed using a one-way ANOVA test and Tukey's multiple comparisons test.

## Quantitative RT-PCR (RT-qPCR) analyses

Total Arabidopsis RNAs from two leaves per individual plant per condition were extracted using Nucleospin RNA plant kit (Macherey Nagel). Five hundred nanograms of DNase-treated RNA were reverse transcribed using qScript cDNA Supermix (Quanta Biosciences). cDNA was then amplified by real time PCR reactions using Takyon SYBR Green Supermix (Eurogentec) and gene specific primers. Expression was normalized to that of the Arabidopsis housekeeping gene Ubiquitin. Sequences of the primers are listed in *Supplementary file 2*.

## McrBC digestion followed by qPCR

McrBC digestion was performed as described previously (*Bond and Baulcombe, 2015*) on 200 ng of DNA. The reaction buffer was composed of 1× NEB Buffer 2, 20 µg BSA, 1 mM GTP in a final volume of 100 µl. Then, each sample was split in two tubes. One hundred nanograms of DNA were digested at 37°C overnight with 10 U of McrBC (New England Biolabs) in one tube, while the enzyme was not added in the control tube. The reactions were inactivated at 65°C during 20 min. Six nanograms of DNA were then amplified by real time PCR reactions using Takyon SYBR Green Supermix

(Eurogentec) and gene-specific primers, and the ratio between digested DNA and undigested DNA was quantified to assess the proportion of methylation (*Bond and Baulcombe, 2015*).

## DAP followed by qPCR analysis

DAP followed by qPCR was performed following a protocol modified from *Bartlett et al., 2017*.

### DNA preparation

For each condition, genomic DNA was extracted from leaves of three independent 5-week-old plants using the CTAB (hexadecyl trimethyl-ammonium bromide) protocol (*Doyle and Doyle, 1990*). Then, 5 µg of genomic DNA were sonicated to obtain DNA fragments of around 200–300 bp long. Sonicated DNA was then precipitated and resuspended in EB buffer (10 mM Tris-HCl, pH = 8.5).

### Protein expression

For each TF or control expressing GST alone, 200 ml cultures of *E. coli* Rosetta strains, carrying pGEX-TW-WRKY or empty vector for the GST alone, at a $OD_{600nm}$ = 0.3–0.6 were induced with 1 mM IPTG during 1 hr at 37°C. Expression of the WRKYs was then confirmed on a Coomassie gel.

### DNA-affinity purification

Bacterial pellets were lysed and sonicated for 10 cycles of 10 s. The supernatants containing expressed proteins were incubated with washed MagneGST beads (Promega) (25 µl per TF and per DNA sample tested) during 1 hr at room temperature with gentle rotation. After five steps of washing with 1× PBS + NP40 (0.005%) and three steps with 1× PBS, beads bound with GST-WRKYs were mixed with 200 ng of sonicated DNAs and incubated 1 hr at room temperature with gentle rotation. Samples were then washed five times with 1× PBS + NP40 (0.005%) and two times with 1× PBS. Beads were then resuspended in 25 µl EB (10 mM Tris-HCl, pH = 8.5) and the samples heated during 10 min at 98°C. Supernatant was kept for further qPCR analysis.

### qPCR following DAP

The equivalent of 1 µl of immunoprecipitated DNA was then amplified by real time PCR reactions using Takyon SYBR Green Supermix (Eurogentec) and gene specific primers, and normalized to the input DNA.

## Northern blot

Accumulation of low molecular weight RNAs was assessed by low molecular weight northern blot analysis as previously described (*Navarro et al., 2008*). Total RNAs were extracted from three independent plants per genotype using TRIzol reagent and stabilized in 50% formamide, and 30 µg of total RNAs were used. To generate specific 32P-radiolabelled probes, regions of 150–300 bp were amplified from the plasmids using gene specific primers listed in *Supplementary file 2*, and the amplicons were labelled by random priming (Prime-a-Gene Labeling System, Promega). U6 was used as an equal loading control.

## GUS staining assay

GUS staining was performed as described previously (*Zervudacki et al., 2018*). The staining was performed on at least 16 individual T1 plants. Two leaves per plant were infiltrated either with mock (water) or with 1 µM flg22 for 24 hr.

## Whole genome bisulfite, mRNA and small RNA sequencing, and bioinformatic datamining

### Small RNA reads processing

Total RNAs from 5-week-old leaves of Col-0, *ros1-3* and *ros1dcl23* mutants were used for small RNA deep-sequencing (two independent biological replicates were used for this analysis). Custom libraries for ~16–40 nt RNAs were constructed and sequenced by Fasteris. We filtered all six libraries based on base-call quality of Q20 (99% base call accuracy). We then selected a subset of read sizes comprised between 16 and 29 nt for further analyses. Reads were mapped to TAIR10 genome using

bowtie (*Langmead et al., 2009*), allowing zero mismatches, and further normalized using a Reads Per Kilobase Million (RPKM) approach. A strong uphill correlation was observed between replicates: 0.79 between Col-0 replicates, 0.81 between *ros1-3* replicates, and 0.77 between *ros1dcl23* replicates.

## Identification of flg22 differentially expressed genes

Total RNAs were extracted from 5-week-old leaves of Col-0 and *ros1-3* mutants syringe-infiltrated with either mock (water) or 1 µM of flg22 for 6 hr, and mRNA-seq experiments further carried out (two independent biological replicates were used for this analysis). Reads were mapped to the TAIR10 genome using TopHat (v2.0.8b) (*Kim et al., 2013*) and allowing two mismatches. In this analysis, reads reporting multiple alignments were tolerated. The score was divided by the total number of hits. Using the stranded mRNA protocol, read counts were reported separately for sense and antisense transcripts. Uninformative annotations were filtered out by keeping only annotations showing at least one count per million (CPM) in two samples simultaneously. After count per million computations, if a gene had only one read reported in only one sample, it was discarded. But if it had one read mapped in at least two samples, it was kept. This step is performed to eliminate non-expressed and non-informative genes that tend to produce noise and ultimately bias internal variance estimations, needed for further steps. Trimmed mean of M-values (TMM) normalization was performed followed by a likelihood ratio test. The resulting p-values were adjusted using Benjamini and Hochberg's approach (*Benjamini and Hochberg, 1995*). An annotation was considered differentially expressed when a log2 fold-change >0 and a p-value <0.05 were observed in 'flg22' vs 'mock' comparison.

## Identification of flg22 differentially expressed genes that are regulated by ROS1

We identified differentially expressed genes separately for Col-0 and *ros1-3* samples in elicited *vs* mock conditions. For each gene, we kept the information of the type of regulation: 'induced' when the p-value after correction is ≤0.05 and the log2 fold-change >0, 'repressed', when the p-value after correction is ≤0.05 and the log2 fold-change <0, or 'none' for the rest of the genes.

In the following analysis, we considered four cases. If there is an induction in Col-0 and there is no regulation in *ros1-3* or the regulation in *ros1-3* is at least two times less important than that of Col-0 (based on fold-change), the gene will be considered as 'less-induced'. Likewise, a 'repressed' gene in Col-0 will be considered as 'less-repressed' if its repression is at least two times less important or if there is no regulation in *ros1-3*.

## Bisulfite-sequencing (Bs-Seq) data mining

Genomic DNAs from 5-week-old leaves of Col-0, *ros1-3* and *ros1dcl23* were extracted using DNeasy plant mini kit (Qiagen) and used for bisulfite sequencing (Bs-Seq) analyses (two independent biological replicates were used for this analysis). We filtered-out low quality reads using trim galore and aligned to TAIR10 genome using Bismark (*Krueger and Andrews, 2011*). Mapping steps were followed by a cytosine methylation call.

## Identification of DMRs

The methodology was conducted as previously reported (*Qian et al., 2012*) and performed for each biological replicate separately. For differentially methylated cytosines (DMCs) call, we kept cytosines having a read coverage of at least two and less than 100 in both the wild-type and mutant samples. A cytosine was considered differentially methylated when the p-value from the two-tailed fisher's test was <0.05, hypermethylated if the level of methylation in the mutant was greater than the WT and hypomethylated in the opposite case. DMRs were retrieved using a sliding windows strategy. We divided the genome into 1000 bp regions with a sliding window of 500 bp. We kept the regions which contained at least 5 DMCs and we redefined the coordinates as the first and last DMCs of that region. When the distance between two regions was less than 100 bp with at least 10 DMCs, they were concatenated into one window. Regions with ambiguous hypermethylated and hypomethylated DMCs were discarded. Finally, we only kept regions that overlapped in both replicates with

at least 10 DMCs and a 10% methylation level shift in one of the three cytosine contexts when comparing Col-0 to *ros1-3*.

## Cross-referencing mRNA-Seq and bisulfite-Seq data

For this analysis, we focused on protein-coding genes having a hyper-DMR 'within 2 Kb' (*i.e.* the region covering the gene-body plus 2 Kb upstream and downstream). Using this method, we selected 2943 protein coding genes. This list was then crossed with the list of 'less-induced genes', which allowed us to retrieve 102 candidates for further analyses.

## Meta-analyses of 'less-induced genes' having a hyper-DMR 'within 2 Kb'

We used deeptools (*Ramírez et al., 2016*) with sliding genomic windows of 100 bp to visualize average methylation levels of 102 candidates in Col-0, *ros1-3* and *ros1dcl23* samples. The rest of 2943 genes having a hyper-DMR within 2 kb were used as control. With the same approach we compared families of small RNAs in all three backgrounds. An analysis was performed for 21-22nt and 23-24nt families separately after a read-per-million normalization.

## GAT analysis conducted on promoter sequence regions from stringent flg22-induced ROS1 targets

To assess whether Arabidopsis TFs could bind to the DNA promoter regions from flg22-induced ROS1 targets, we first selected a subset of 30 stringent ROS1 targets (exhibiting dense hyperDMR in their promoter regions and which are strongly impaired in their induction in the *ros1*-elicited mutant background). We used the Genomic Association Tester (GAT) (*Heger et al., 2013*) to find significant association between DMRs and experimentally validated TF binding sites from previously published DAP-seq data (*O'Malley et al., 2016*). A TF was kept when its p-value and fold-change of association were respectively <0.05 and >1. K means analysis was performed to see which genes group together when we take the WRKY binding profile as criterion. The distance tree displayed on top of the heatmap was obtained using the GAT scores to perform a hierarchical clustering; here we computed a one minus Pearson correlation to draw distances. The size of the circle represents the log2-fold-change of GAT analysis (a higher log2-fold-change, corresponding to a bigger circle, means the TF binds more often to this particular promoter than in 1000 simulations of randomly assigned genomic intervals of the same size).

## Acknowledgements

We thank Pr. Imre Somssich for providing the pGEX-TM-WRKY40 plasmid and the ChIP-seq raw datasets ahead of publication, Pr. Robert Fischer for providing the *ros1-3* and *ros1-4* alleles, Pr. Sheng Yang He for providing the GFP-tagged *Pto* DC3000 strain, Dr. Florence Jay for her contribution to the selection of the *ros1-3 dcl2-1 dcl3-1* mutants, and all the members of the Navarro lab for their inputs, discussions, and critical reading of the manuscript.

## Additional information

### Funding

| Funder | Grant reference number | Author |
|---|---|---|
| H2020 European Research Council | 281749 - Silencing & Immunity | Lionel Navarro |
| Agence Nationale de la Recherche | ANR-18-CE20-0020 - NEPHRON | Lionel Navarro |
| H2020 Marie Skłodowska-Curie Actions | EU Project 661715 - BASILA | Thierry Halter |
| Agence Nationale de la Recherche | ANR-10-IDEX-0001-02PSL | Jingyu Wang Meenu Singla Rastogi |
| Agence Nationale de la Recherche | ANR-10-LABEX-54 MEMOLIFE | Jingyu Wang Meenu Singla Rastogi |

| Fondation Pierre-Gilles de Gennes pour la recherche | Post-doctoral fellowship | Thierry Halter |

The funders had no role in study design, data collection and interpretation, or the decision to submit the work for publication.

## Author contributions

Thierry Halter, Funding acquisition, Validation, Investigation, Visualization, Writing - original draft; Jingyu Wang, Meenu Singla Rastogi, Funding acquisition, Validation, Investigation; Delase Amesefe, Software, Formal analysis, Investigation, Visualization, Writing - review and editing; Emmanuelle Lastrucci, Software, Formal analysis, Investigation; Magali Charvin, Validation, Investigation, Writing - review and editing; Lionel Navarro, Conceptualization, Supervision, Funding acquisition, Project administration, Writing -original draft, review and editing

## Author ORCIDs

Thierry Halter https://orcid.org/0000-0002-4050-5981
Magali Charvin https://orcid.org/0000-0003-1165-4336
Lionel Navarro https://orcid.org/0000-0003-1083-9478

## Decision letter and Author response

Decision letter https://doi.org/10.7554/eLife.62994.sa1
Author response https://doi.org/10.7554/eLife.62994.sa2

# Additional files

## Supplementary files

- Supplementary file 1. DMRs position of 102 less-induced and 115 less-repressed ROS1 targets.
- Supplementary file 2. Primers used in this study.
- Transparent reporting form

## Data availability

All data generated or analysed during this study are included in the manuscript and supporting files. Source data files have been provided for all main figures and for figure supplements. Sequencing data have been deposited in SRA under the accession code SRP133028.

The following dataset was generated:

| Author(s) | Year | Dataset title | Dataset URL | Database and Identifier |
|---|---|---|---|---|
| Halter T, Wang J, Amesefe D, Lastrucci E, Charvin M, Rastogi MS, Navarro L | 2020 | The Arabidopsis demethylase ROS1 cis-regulates defense genes by erasing DNA methylation at promoter-regulatory regions | https://www.ncbi.nlm.nih.gov/sra/?term=srp133028 | NCBI Sequence Read Archive, SRP133028 |

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
