## [Decision Letter]

**Acceptance summary:**

This manuscript describes a carefully conducted study about the regulatory function of DNA (de)methylation in Arabidopsis defence gene expression. The authors show that DNA demethylation positively controls defence-related expression of a disease resistance gene, thereby contributing to basal resistance to bacterial infection. This paper clearly establishes a causal relationship between DNA methylation, transcription factor binding, and pathogen-responsive gene expression in plants.

**Decision letter after peer review:**

Thank you for submitting your article "The Arabidopsis demethylase ROS1 *cis*-regulates defense genes by erasing DNA methylation at promoter-regulatory regions" for consideration by *eLife*. Your article has been reviewed by three peer reviewers, including Daniel Zilberman as the Reviewing Editor and Reviewer #1, and the evaluation has been overseen by Detlef Weigel as the Senior Editor. The following individuals involved in review of your submission have agreed to reveal their identity: Jian-Kang Zhu (Reviewer #2); Jurriaan Ton (Reviewer #3).

The Reviewing Editor has drafted this decision to help you prepare a revised submission.

Summary:

This interesting manuscript describes a carefully conducted study about the regulatory function of DNA (de)methylation in WRKY-dependent defence gene expression in Arabidopsis. The work follows a previous pioneering study by this group from 2013 (Yu et al., 2013), which established a role for RdDM and ROS1-dependent DNA de-methyaltion in Arabidopsis resistance against *P. syringae* pv. tomato (Pst). The current study focuses largely on two defence-related genes, *RMG1* and *RLP43*, which are negatively controlled by ROS1. The authors show that ROS1-dependent DNA demethylation of a RdDM-targeted AtREP4 element in the *RMG1* promoter positively controls defence-related expression of this resistance gene, thereby contributing to basal resistance against Pst. Subsequently, the authors selected genes displaying repressed induction by the bacterial PAMP flg22 and increased DNA methylation in the *ros1-3* mutant. Using the epicistrome dataset of the O'Malley et al., 2016 study, a subset of these Flg22-inducible genes was analysed for methylation-sensitive TF binding sites, revealing enrichment of WRKY-binding W-box elements. In the final part of the manuscript, the authors employ state-of-the-art DAP-qPCR and artificial siRNA-directed re-methylation techniques to establish a causal relationship between ROS1-dependent DNA methylation of the W-box in the *RLP43* gene promoter, its binding to WRKY18 and WRKY40 TFs, and Flg22-responsive expression.

The authors generate compelling evidence that ROS1-dependent DNA demethylation at some defence gene promoters is required for WRKY-dependent induction following PAMP perception. The manuscript is clearly written, follows a logical narrative, and includes all details and data necessary to repeat experiments and re-analyze data.

Essential revisions:

1) Subsection “ROS1 directs promoter demethylation of flg22-induced ROS1 targets, mostly by antagonizing DCL2 and/or DCL3 functions”: The authors state that there were 219 Flg22-responsive candidate genes that are associated with regions that are hyper-methylated in the *ros1-3* mutant, of which 102 were "less- induced" and 115 were "less-repressed". These numbers don't add up though: 102 + 115 = 217. Furthermore, the hyperDMRs at these 219 (or 217?) genes are reported to be within the gene bodies plus 2Kb upstream/downstream regions. It would be helpful if the authors could show these data for the 115 less-repressed and 102 less-induced genes separately, and indicate the distribution of ROS-dependent methylation between gene bodies, 2 Kb upstream sequences and 2 Kb downstream sequences.

2) Related to the above point, could the authors comment on how ROS1-dependent DNA demethylation may control the 115 genes showing less flg22-induced repression in the *ros1-3* mutant? For these genes, one would expect that ROS1-dependent DNA demethylation facilitates binding of repressive TFs. Are the promoters of these Flg22-repressed genes also associated with methylation-sensitive W box elements?

3) Using their DAP-qPCR system, the authors convincingly demonstrate in Figure 5 that WRKY18-GST and WRKY40-GST have lower affinity to the methylated W-box within the *RLP43* promoter of the *ros1-3* mutant. Is this also true for the W-box element in the promoter of the *RMG1* gene?

---

## [Author Response]

Essential revisions:1) Subsection “ROS1 directs promoter demethylation of flg22-induced ROS1 targets, mostly by antagonizing DCL2 and/or DCL3 functions”: The authors state that there were 219 Flg22-responsive candidate genes that are associated with regions that are hyper-methylated in the ros1-3 mutant, of which 102 were "less- induced" and 115 were "less-repressed". These numbers don't add up though: 102 + 115 = 217. Furthermore, the hyperDMRs at these 219 (or 217?) genes are reported to be within the gene bodies plus 2Kb upstream/downstream regions. It would be helpful if the authors could show these data for the 115 less-repressed and 102 less-induced genes separately, and indicate the distribution of ROS-dependent methylation between gene bodies, 2 Kb upstream sequences and 2 Kb downstream sequences.

The number of flg22-responsive candidate genes is indeed 217. This mistake has been corrected. To address reviewers’ suggestions, we have added the distribution of ROS1-dependent demethylation regions between gene bodies, 2 Kb upstream sequences and 2 Kb downstream sequences. Furthermore, we now provide the hyperDMRs location for each ROS1 target from the 115 less-repressed and 102 less-induced gene sets.

This information has been added in the Results section as well as in Supplementary file 1:

“Among them, 102 less-induced (promoter regions: 44 DMRs; gene bodies: 33; downstream regions: 25) and 115 less-repressed (promoter regions: 67 DMRs; gene bodies: 27; downstream regions: 21) genes were recovered (Figure 3A, Supplementary file 1).”

2) Related to the above point, could the authors comment on how ROS1-dependent DNA demethylation may control the 115 genes showing less flg22-induced repression in the ros1-3 mutant? For these genes, one would expect that ROS1-dependent DNA demethylation facilitates binding of repressive TFs. Are the promoters of these Flg22-repressed genes also associated with methylation-sensitive W box elements?

To address this point, we performed a GAT analysis on 28 less-repressed ROS1 targets, which exhibit dense hyperDMRs at their promoter regions in *ros1*-*3* mutants. By doing this analysis, we found that WRKYs enrichment was detected at only 2 candidate demethylated promoter regions. These data indicate that the flg22-triggered repression of these genes occurs mostly independently of WRKYs DNA binding. In addition, we noticed that other classes of TFs can bind to the demethylated promoter regions of flg22-repressed ROS1 targets. As proposed by the reviewers, ROS1 might direct demethylation of these promoters to facilitate binding of various repressive TFs, thereby ensuring a proper repression of these genes during PTI. We have now included a new figure presenting these results (see Figure 4—figure supplement 2). Furthermore, we have added the following sentences in the Results and Discussion sections of the revised manuscript to describe and discuss these new results:

“By contrast, only 2 out of 28 less-repressed stringent targets displayed WRKY TFs enrichment at their demethylated promoter regions (Figure 4—figure supplement 2)”.

“It is noteworthy that the proposed mechanism might also hold true for other TF families, which exhibit in vitro binding at the demethylated promoter regions of flg22-induced ROS1 targets (Figure 4—figure supplement 1). […] Consistent with this hypothesis, different classes of TFs were found to preferentially bind to the demethylated promoter regions of the flg22-repressed ROS1 targets (Figure 4—figure supplement 2).”

3) Using their DAP-qPCR system, the authors convincingly demonstrate in Figure 5 that WRKY18-GST and WRKY40-GST have lower affinity to the methylated W-box within the RLP43 promoter of the ros1-3 mutant. Is this also true for the W-box element in the promoter of the RMG1 gene?

The available DAP-seq datasets revealed a weak in vitro binding of WRKY TFs at the demethylated region of *RMG1*, which was difficult to reproducibly detect using DAP-qPCR analysis. Furthermore, the presence of W-box *cis*-elements in the close vicinity of the *RMG1* hyperDMR is expected to complexify the interpretation of DAP-qPCR results as the primers flanking this region also amplify W-box sequences that are not subjected to ROS1-dependent demethylation. By contrast, the DAP-seq datasets at the *RLP43* demethylated promoter region revealed robust in vitro binding of several WRKY TFs, which was convincingly detectable by DAP-qPCR analysis. Furthermore, only a single –and functional– W-box *cis*-regulatory element is present right in the middle of the *RLP43* hyperDMR, which is devoid of W-box in its surrounding. The *RLP43* promoter is therefore the most appropriate candidate to investigate the relevance of demethylation in WRKY TFs binding using our DAP-qPCR system. In the revised manuscript, we are now providing an explanation for why we made this choice. In the future, we intend to conduct DAP-seq analyses in Col-0 versus *ros1* mutants, which should address reviewers’ comments at *RMG1* but also at other ROS1 targets showing similar features. This approach will also determine the extent to which ROS1-directed demethylation facilitate WRKY TFs binding at the whole genome level.

The following sentence has been added in the Results section of our revised manuscript:

“The rationale for using *RLP43* promoter for this specific analysis was based on (1) previous DAP-seq and ChIP-seq data showing robust binding of different WRKY TFs at the ROS1-directed demethylated *RLP43* promoter (Figure 4A-C, Figure 4—figure supplement 1), (2), the presence of a single W-box *cis*-element located right in the middle of this genomic region (Figure 4B, C) and, (3) the central role of this W-box in the flg22-responsiveness of the *RLP43* promoter (Figure 4D-F).”